# Large-scale deorphanization of *Nematostella vectensis* neuropeptide G protein-coupled receptors supports the independent expansion of bilaterian and cnidarian peptidergic systems

Daniel Thiel[1]*[†], Luis Alfonso Yañez Guerra[1†‡], Amanda Kieswetter[2], Alison G Cole[3], Liesbet Temmerman[2], Ulrich Technau[3], Gáspár Jékely[1,4]*

[1]Living Systems Institute, University of Exeter, Exeter, United Kingdom; [2]Animal Physiology & Neurobiology, Department of Biology, University of Leuven, Leuven, Belgium; [3]Department of Neurosciences and Developmental Biology, Faculty of Life Sciences, University of Vienna, Vienna, Austria; [4]Centre for Organismal Studies (COS), Heidelberg University, Heidelberg, Germany

*For correspondence:
d.thiel@exeter.ac.uk (DT);
gaspar.jekely@cos.uni-heidelberg.de (GJ)

[†]These authors contributed equally to this work

Present address: [‡]School of Biology, Institute for Life Sciences, University of Southampton, Southampton, United Kingdom

**Abstract** Neuropeptides are ancient signaling molecules in animals but only few peptide receptors are known outside bilaterians. Cnidarians possess a large number of G protein-coupled receptors (GPCRs) – the most common receptors of bilaterian neuropeptides – but most of these remain orphan with no known ligands. We searched for neuropeptides in the sea anemone *Nematostella vectensis* and created a library of 64 peptides derived from 33 precursors. In a large-scale pharmacological screen with these peptides and 161 *N. vectensis* GPCRs, we identified 31 receptors specifically activated by 1 to 3 of 14 peptides. Mapping GPCR and neuropeptide expression to single-cell sequencing data revealed how cnidarian tissues are extensively connected by multilayer peptidergic networks. Phylogenetic analysis identified no direct orthology to bilaterian peptidergic systems and supports the independent expansion of neuropeptide signaling in cnidarians from a few ancestral peptide-receptor pairs.

## eLife assessment

This work identifies cnidarian neuropeptides and pairs them to their GPCR, then shows that neuropeptide signaling systems have evolved and diversified independently in cnidarians and bilaterians. Neuropeptide-receptor partners were experimentally identified using established and widely used methodologies including single cell mapping, providing **compelling** evidence for the conclusions of the paper. This impressive accomplishment provides **fundamental** new insights into the evolution of neuropeptide signaling systems and will be of broad interest to neurobiologists and evolution of development researchers.

## Introduction

The origin of neuropeptides predates the emergence of neurons and it is believed that these signaling molecules have been utilized in the most ancestral nervous systems (*Jékely, 2021*; *Moroz et al., 2021*; *Yañez-Guerra et al., 2022*). Neuropeptide-like molecules occur in all major animal clades with neurons, the Bilateria, Cnidaria, and Ctenophora, and even in the neuron-less Placozoa and Porifera

(*Hayakawa et al., 2022*; *Koch and Grimmelikhuijzen, 2019*; *Nikitin, 2015*; *Sachkova et al., 2021*; *Yañez-Guerra et al., 2022*). However, the deep relationships of animal neuropeptidergic systems, in particular between Bilateria and non-bilaterians, have remained elusive.

Most mature bioactive neuropeptides are 3–20 amino acids long and derive from longer proneuropeptide precursors through cleavage and other post-translational modifications. The same set of enzymes are involved in proneuropeptide processing across animals. These include prohormone convertases, which recognize dibasic cleavage sites flanking the active peptides, and peptidyl-glycine alpha-amidating monooxygenase, which converts C-terminal glycine residues to amide groups (*Chufán et al., 2009*; *Seidah, 2011*). Proneuropeptide precursors can contain a single peptide or multiple copies of identical or divergent peptide sequences separated by cleavage sites. The often repetitive structure and the presence of short active sequences interspersed with less-constrained interpeptide regions allow propeptide sequences to evolve relatively rapidly. Consequently, with increasing evolutionary distances it often becomes hard or impossible to recognize orthologous proneuropeptide sequences. Even within Bilateria, the orthology relationship of many neuropeptide families between protostomes and deuterostomes have only been recognized due to the orthology of their receptors (e.g. vertebrate orexin and insect allatotropin) (*Alzugaray et al., 2019*; *Elphick et al., 2018*; *Mirabeau and Joly, 2013*; *Semmens et al., 2015*).

Most neuropeptides signal through G protein-coupled receptors (GPCRs), which are larger seven-transmembrane proteins that show a slower evolutionary rate than proneuropeptide precursors. The cases where the evolution of both the proneuropeptides and their receptors could be reconstructed revealed a strong pattern of coevolution between receptor and ligand (*Elphick et al., 2018*; *Escudero Castelán et al., 2022*; *Grimmelikhuijzen and Hauser, 2012*; *Jékely, 2013*; *Mirabeau and Joly, 2013*; *Yañez-Guerra and Elphick, 2020*). This coevolutionary pattern has been used to also trace the evolution of peptide families where the ligands are too divergent to retain phylogenetic signal. These analyses revealed more than 30 conserved peptidergic signaling systems across bilaterians (*Elphick et al., 2018*; *Jékely, 2013*; *Mirabeau and Joly, 2013*; *Thiel et al., 2021*; *Yañez-Guerra et al., 2022*).

The nervous system of cnidarians has long been known to be strongly peptidergic and proneuropeptides have been found across all major cnidarian clades (*Koch et al., 2021*; *Koch and Grimmelikhuijzen, 2020*). Most cnidarian neuropeptides are short amidated peptides, resembling those found in bilaterians. Genomic comparisons revealed three proneuropeptide families that were present in the cnidarian stem lineage. All three give rise to short amidated neuropeptides: GLWamides, GRFamides, and PRXamides (with 'X' representing a variable amino acid residue) (*Koch et al., 2021*; *Koch and Grimmelikhuijzen, 2020*). Other neuropeptides are more specific to certain cnidarian groups such as QITRFamide and HIRamide to Hexacorallia or LRWamides to Anthozoa (*Koch and Grimmelikhuijzen, 2020*). Peptides that are sufficiently similar to be considered orthologous between bilaterians and cnidarians are restricted to a few atypical neuropeptides. These include insulin-related peptides, glycoprotein-hormone-related peptides, trunk-related peptides, nesfatin, and phoenixin (*de Oliveira et al., 2019*; *Roch and Sherwood, 2014*; *Yañez-Guerra et al., 2022*). The more common short amidated peptides, however, have no clear orthologs in bilaterians. Similarity is at most restricted to one or two C-terminal residues, such as between RFamide or Wamide peptides (*Walker et al., 2009*; *Williams, 2020*; *Jékely, 2013*). Sometimes this has been interpreted as evidence of common origin but there is no other evidence supporting potential orthologies for these cnidarian peptides and their bilaterian counterparts. Receptors for most cnidarian neuropeptides are still unknown, with two exceptions. These include receptors for *Hydra vulgaris* RFamide peptides and a receptor for a PRXamide maturation-inducing hormone (MIH) in the hydrozoan *Clytia hemisphaerica* (*Quiroga Artigas et al., 2020*; *Assmann et al., 2014*). *Hydra* RFamide peptides activate heterotrimeric peptide-gated ion channels belonging to the DEG/ENaC family and distantly related to bilaterian RFamide- and Wamide-gated ion channels (*Dandamudi et al., 2022*; *Elkhatib et al., 2022*; *Gründer et al., 2022*). The *Clytia* MIH receptor is a class A GPCR, which together with related cnidarian GPCRs shows a many-to-many ortholog relationship to a range of bilaterian neuropeptide GPCR families that also contain receptors for RFamide-like neuropeptides (*Quiroga Artigas et al., 2020*).

Cnidarian genomes can encode a large number of class A GPCRs. The genome of the sea anemone *Nematostella vectensis*, for example, contains over 1000 GPCR genes (*Krishnan and Schiöth, 2015*). Earlier analyses suggested direct orthologous relationships between several cnidarian and bilaterian neuropeptide GPCRs such as orexin/allatotropin, somatostatin or neuropeptide Y receptors, besides

others (*Alzugaray et al., 2013*; *Anctil, 2009*; *Krishnan and Schiöth, 2015*). However, these results were only based on either BLAST similarity or limited phylogenetic analyses. More recent and comprehensive phylogenies suggest that cnidarian neuropeptide GPCRs are more closely related to each other than to bilaterian neuropeptide GPCRs and only show many-to-many, or few-to-many orthology with bilaterian receptors (*Quiroga Artigas et al., 2020*; *Hauser et al., 2022*; *Thiel et al., 2018*). However, a comprehensive phylogenetic analysis of cnidarian GPCRs is still lacking. This, together with the paucity of experimentally characterized receptors, leaves our understanding of the evolution of eumetazoan peptidergic systems fragmentary.

Here, in a large-scale bioinformatic analysis, we map the global sequence diversity of metazoan class A GPCRs to identify neuropeptide GPCR candidates in cnidarians. We then use mass spectrometry and bioinformatics to compile a library of predicted *N. vectensis* neuropeptides. In a combinatorial pharmacological ligand-receptor assay, we test our peptide library against selected *N. vectensis* GPCRs and identify 31 neuropeptide receptors. By phylogenetic analysis, we reconstruct ancestral cnidarian GPCR A systems and their relationship to bilaterian systems. Finally, we map proneuropeptide and GPCR expression to a single-cell RNAseq dataset (*Cole et al., 2024*) to analyze tissue-level

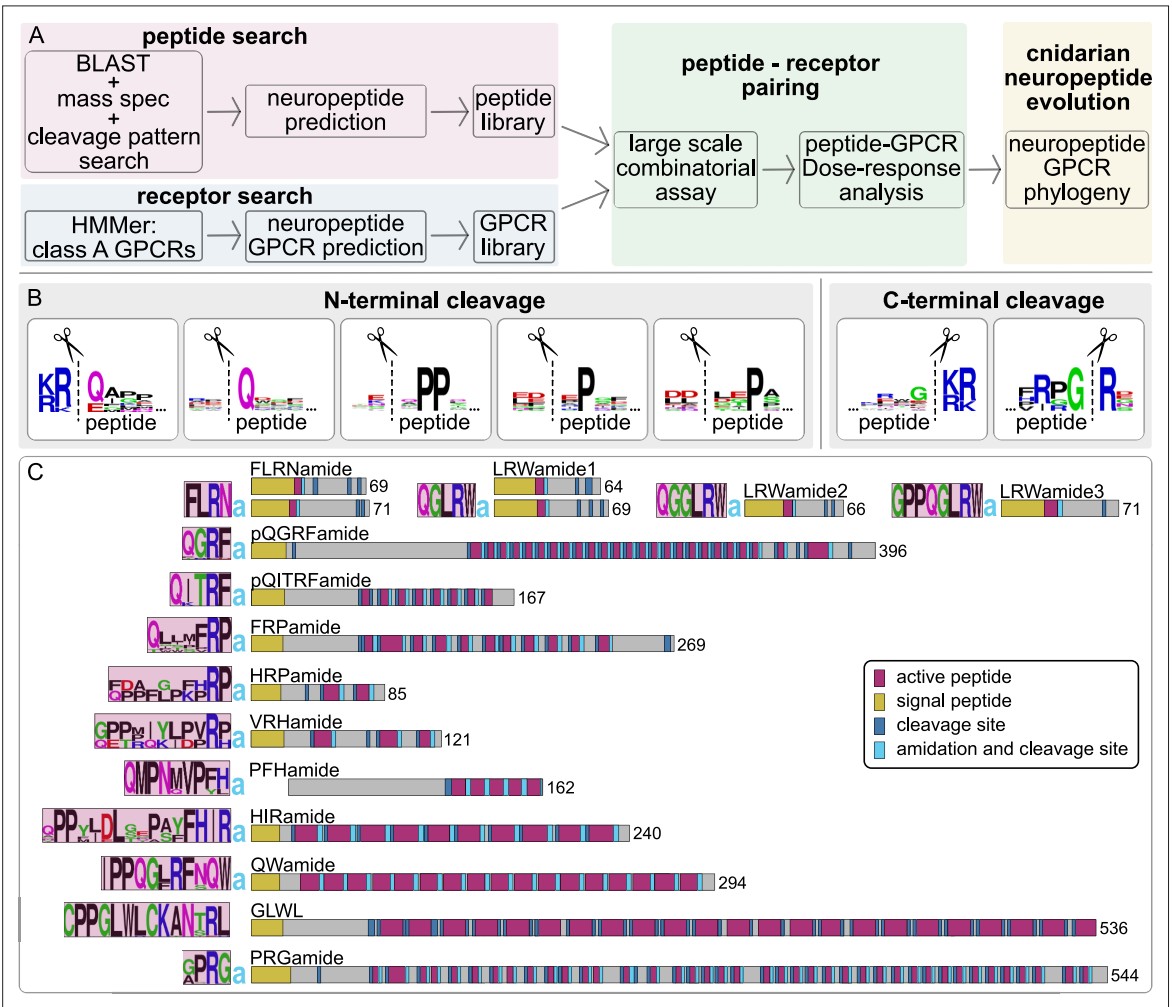

**Figure 1.** Identification of *N. vectensis* neuropeptides. (**A**) Pipeline to identify neuropeptides and their receptors and to reconstruct the evolution of cnidarian peptidergic signaling. (**B**) Peptide sequence logos of N-terminal and C-terminal peptide cleavage sites based on peptides detected by LC-MS/MS. Cleavage occurs at the dashed lines. (**C**) *N. vectensis* neuropeptide precursor schemes of peptides for which we identified a receptor, with sequence logos of the encoded peptide(s) on the left and length of precursor on the right. a=amide.

The online version of this article includes the following figure supplement(s) for figure 1:

**Figure supplement 1.** Mass spectrometry pipeline.

peptidergic signaling in *N. vectensis*. Our results reveal extensive peptidergic networks in *N. vectensis* and support the independent expansion of neuropeptide signaling in Cnidaria and Bilateria.

## Results and discussion
### Creation of a *Nematostella* neuropeptide library

To obtain a comprehensive library of endogenous neuropeptides in *N. vectensis,* we extended the list of known proneuropeptides (*Hayakawa et al., 2019*; *Koch and Grimmelikhuijzen, 2020*) by mass spectrometry and bioinformatic screening (*Figure 1A*). We first screened a *N. vectensis* transcriptome collection for sequences encoding a signal peptide. This predicted secretome was filtered with regular expressions to detect sequences with the repetitive dibasic cleavage sites (K and R in any combination) and amidation sites, using a custom script from a previous publication (*Thiel et al., 2021*). In addition, we used standard BLAST searches using known cnidarian neuropeptide precursors. In parallel, we analyzed methanolic extracts from *N. vectensis* larval, juvenile, and adult tissue by mass spectroscopy (LC-MS/MS) (*Figure 1—figure supplement 1*). The LC-MS/MS analysis confirmed the existence of peptides predicted from various precursor sequences (*Supplementary files 1 and 2*). It also confirmed the existence of two different precursors that encode the FLRNamide peptide and two different precursors that encode the LRWamide1 peptide, by identifying non-amidated peptides that are encoded between cleavage sites next to the name-giving peptides LRWamide and FLRNamide (*Figure 1C*, *Supplementary files 1 and 2*, *Koch and Grimmelikhuijzen, 2020*).

MS confirmed the occurrence of N-terminal peptide cleavage at dibasic KR-x, RR-x, and KK-x sites, with 'x' indicating the N-terminal amino acid of the resulting active peptide (*Figure 1B*, *Supplementary file 1*). Cleavage at such dibasic sites is typical for bilaterian precursors (*Southey et al., 2008*; *Veenstra, 2000*). In *Nematostella*, the cleavage site is often followed by a Q residue as the most N-terminal amino acid of the peptide (*Figure 1B*). Cleavage at dibasic sites, however, does not occur in all precursors. Our MS data showed that some peptides are processed by cleavage N-terminal to a Q residue without any basic residue but the cleavage site is instead often accompanied by an acidic (D or E) residue 1 or 2 positions N-terminal of the cleavage site (*Figure 1B*, *Supplementary file 1*). Another motif includes two proline residues in positions 2 and 3 from the N-terminus of the peptide, and in some cases a single proline in either the second or third position (*Figure 1B*, *Supplementary file 1*). These proline-related cleavage sites are also often accompanied by an acidic residue flanking the peptide N-terminal of the cleavage site. Such non-dibasic cleavage sites have previously been proposed for Cnidaria (*Hayakawa et al., 2022*; *Hayakawa et al., 2019*; *Koch and Grimmelikhuijzen, 2019*). Some precursors, such as the one for HIRamide, also showed alternative N-terminal cleavage of the same peptide copy, resulting in different versions of the same peptide (*Figure 1—figure supplement 1*, *Supplementary file 1*), indicating stepwise cleavage or controlled degradation of peptides. C-terminal peptide cleavage occurs at dibasic x-KR, x-RR, x-KK and in some cases at alternative monobasic Rxx-R sites, with a second basic amino acid 3 positions N-terminal of the cleavage site (*Figure 1B*, *Supplementary file 1*). This is similar to the C-terminal cleavage found in bilaterian propeptides (*Southey et al., 2008*; *Veenstra, 2000*). In many neuropeptides, we could also confirm peptide alpha-amidation by the conversion of a C-terminal glycine residue to an amide group.

Based on the MS data, we included the additional, non-dibasic N-terminal cleavage sites into our script that uses regular expressions to search for repetitive cleavage sites (*Thiel et al., 2024*) and re-screened the predicted secretome. With our combined approach, we could identify novel neuropeptide precursors, verify the processing of known neuropeptide precursors, and refine cleavage site predictions. We used this information to prepare a list of 33 *N. vectensis* proneuropeptides, excluding potential isoforms of the same precursor but including two potential paralogs of the FLRNamide, LRWamide1, and pyrQITRFamide peptide precursors (*Supplementary file 2*). Our screen complements the list of known neuropeptides in *N. vectensis* (*Hayakawa et al., 2022*; *Hayakawa et al., 2019*; *Koch and Grimmelikhuijzen, 2020*) with 15 new neuropeptide precursors. However, our list did not contain the recently identified GGYamide, GTEamide, and IVLamide peptides (*Hayakawa et al., 2022*) or bursicon- and insulin-like peptides.

We inspected all precursors individually and predicted signal peptides, cleavage sites, and amidation sites (*Figure 1C*, *Supplementary file 2*). Based on our cleavage-site predictions, we then compiled a library of 64 synthetic *Nematostella* neuropeptides, including different versions of peptides from the

same precursors and alternatively cleaved peptides that differ in the length of their N-terminal region (*Supplementary file 2* and *Supplementary file 3*).

## Analysis of metazoan class A GPCRs and selection of *N. vectensis* neuropeptide-receptor candidates

To identify neuropeptide-GPCR candidates in *N. vectensis*, we focused on class A GPCRs representing the main type of neuropeptide receptors in bilaterians (*Jékely, 2013*; *Mirabeau and Joly, 2013*). We first aimed to get an overview of the diversity of class A GPCRs across metazoans and screened transcriptomes of nine cnidarian, six bilaterian, two placozoan, three ctenophore, and five sponge species for these receptors (*Supplementary files 5 and 6*). The number of GPCRs across Metazoa varies by species and seems not to correlate with phylogenetic affiliation (*Figure 2A*). From the combined *N. vectensis* transcriptomes (see *Supplementary file 5*), we initially detected a total of 1061 class A GPCR sequences. We analyzed these together with over 7000 GPCRs retrieved from the other species. As a reference, we used the experimentally confirmed and annotated human, *Drosophila melanogaster*, *Caenorhabditis elegans*, and *Platynereis dumerilii* sequences. We carried out a sequence-similarity-based clustering analysis of all sequences. This revealed only a few well-connected clusters of receptors conserved across all major metazoan lineages (*Figure 2—figure supplement 1*). These include the leucine-rich-repeat containing (lrrc) GPCRs, which are receptors for larger peptides such as bursicons and related glycoprotein hormones or for relaxins and related insulin-like peptides. Other well-connected GPCR clusters were more restricted to the individual phylogenetic groups. Mammalian olfactory receptors represent such a GPCR cluster with limited taxonomic breadth (*Figure 2—figure supplement 1*).

For a more detailed analysis, we then reduced our sampled species to cnidarians, bilaterians with experimentally confirmed GPCRs, *Petromyzon marinus*, and two placozoan species (*Figure 2B*). We re-ran the clustering analysis and filtered out non-connected single sequences, species-specific cnidarian clusters, and unconnected clusters only containing bilaterian or placozoan sequences. The two major clusters in our GPCR map contain bilaterian and cnidarian sequences, interspersed with only a few placozoan GPCRs (*Figure 2C*). In both major clusters, cnidarian and bilaterian sequences form separate subclusters, rather than cnidarian GPCRs being interspersed with bilaterian GPCRs. One of these clusters contains bilaterian GPCRs for low-molecular-weight neurotransmitters, including monoamines, trace amines, acetylcholine, melatonin, and ATP. This cluster is weakly connected to the opsins. The second major cluster contains most bilaterian neuropeptide receptors (*Figure 2C* – neuropeptide II, III, gamma) as well as the bilaterian-specific chemokine, puringeric-P2Y, and fatty acid receptors. There is a second cluster of bilaterian neuropeptide receptors weakly connected to this main cluster (*Figure 2C* – neuropeptide I), which has only few cnidarian sequences connected to it. Based on this cluster analysis, we selected 161 *Nematostella* GPCR sequences. We focused on full-length GPCRs that are associated with the bilaterian neuropeptide GPCR clusters. In addition, we chose candidates from non-connected clusters that are uncharacterized but ancestral to all cnidarians, except the leucine-rich repeat-containing GPCRs, which are known to be activated by insulin-related and glycoprotein-hormone-related peptides in bilaterians.

## Deorphanization of 31 *Nematostella* neuropeptide receptors

To experimentally identify *Nematostella* neuropeptide GPCRs, we tested our selection of 161 *N. vectensis* GPCRs in a pharmacological assay for activation by peptides from our peptide library (*Figure 3A*, *Supplementary files 3 and 8*). We expressed the candidate GPCRs in mammalian cells, together with a promiscuous Gqi protein and a luminescent G5A reporter (*Figure 3A*, *Supplementary file 7*). The peptides were separated into different mixes (*Supplementary file 3*), which were then tested on each GPCR (*Supplementary file 4*). Receptor-mix combinations that gave a positive signal were further resolved by testing the individual peptides of the mix to identify the activating ligand (*Supplementary file 4*). Peptides activating a receptor were then tested at different concentrations to record dose-response curves and determine $EC_{50}$ values for each peptide-receptor pair (*Supplementary file 9*).

In this screen, we identified 31 *N. vectensis* GPCRs activated mostly in the nanomolar range by peptides from 14 different precursors (*Figures 1C, 3B, C, and D*; *Supplementary file 10*). The neuropeptides GLWL, PFHamide, VRHamide, and QWamide each activate a different, single receptor

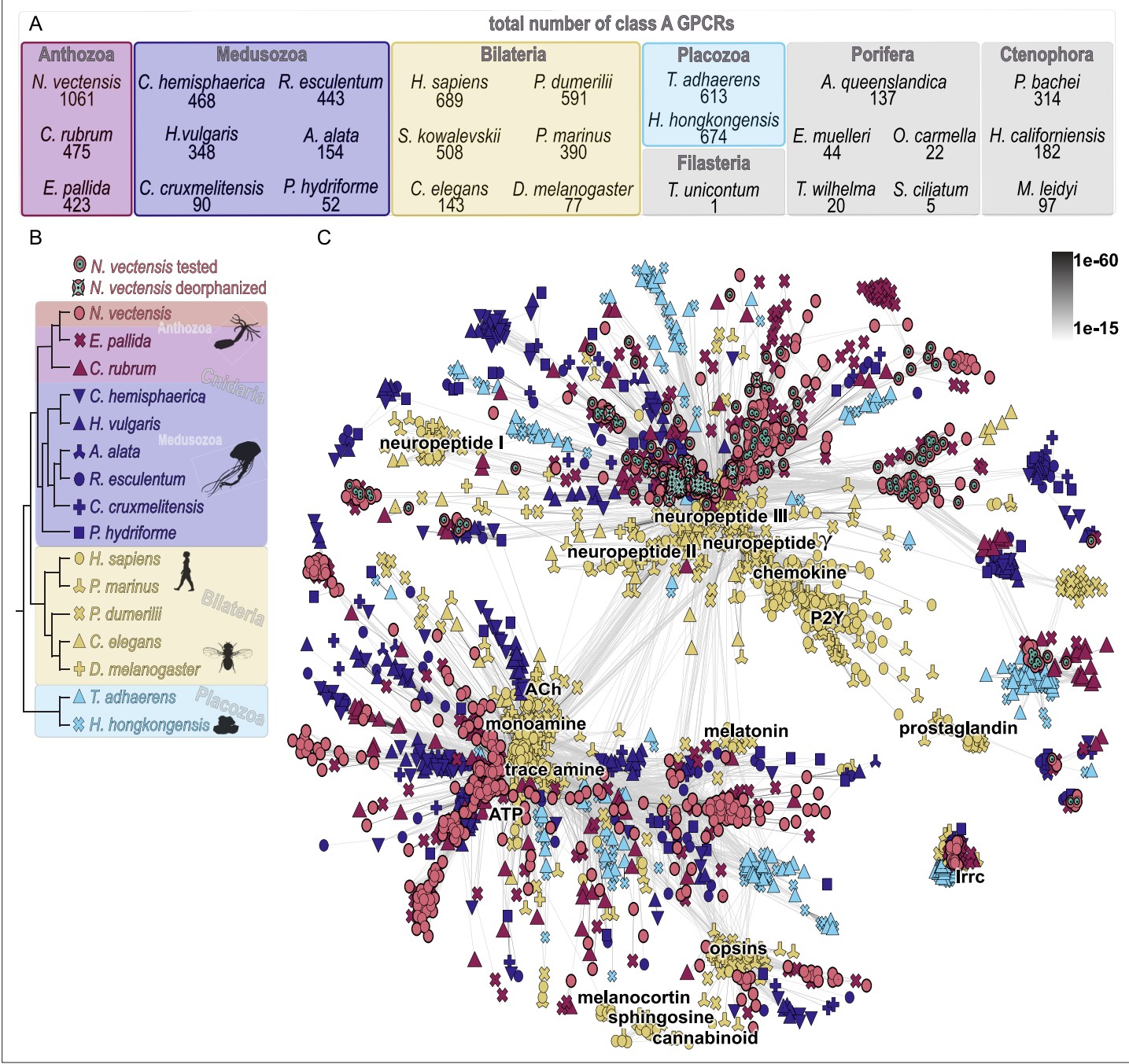

**Figure 2.** Cluster map of selected class A G protein-coupled receptors (GPCRs). (**A**) Number of class A GPCRs identified by HMMer search in the different investigated species. (**B**) Relationship of species used for cluster analysis in C. (**C**) Cluster analysis of major class A GPCR groups from cnidarian, bilaterian, and placozoan species. Each dot represents a GPCR sequence with color-coding and symbols according to the phylogeny in B. Connecting lines between single sequences show similarity with p-values indicated in the top right. Cluster annotations are based on deorphanized bilaterian class A GPCRs. Abbreviations in C: ACh = acetylcholine, lrrc = leucine-rich-repeat containing, P2Y=purinergic P2Y receptor. Silhouette images in B were taken from phylopic.org.

The online version of this article includes the following source data and figure supplement(s) for figure 2:

**Source data 1.** Raw cluster analysis CLANS file.

**Figure supplement 1.** Cluster map of metazoan class A G protein-coupled receptors (GPCRs).

**Figure supplement 1—source data 1.** Raw cluster analysis CLANS file.

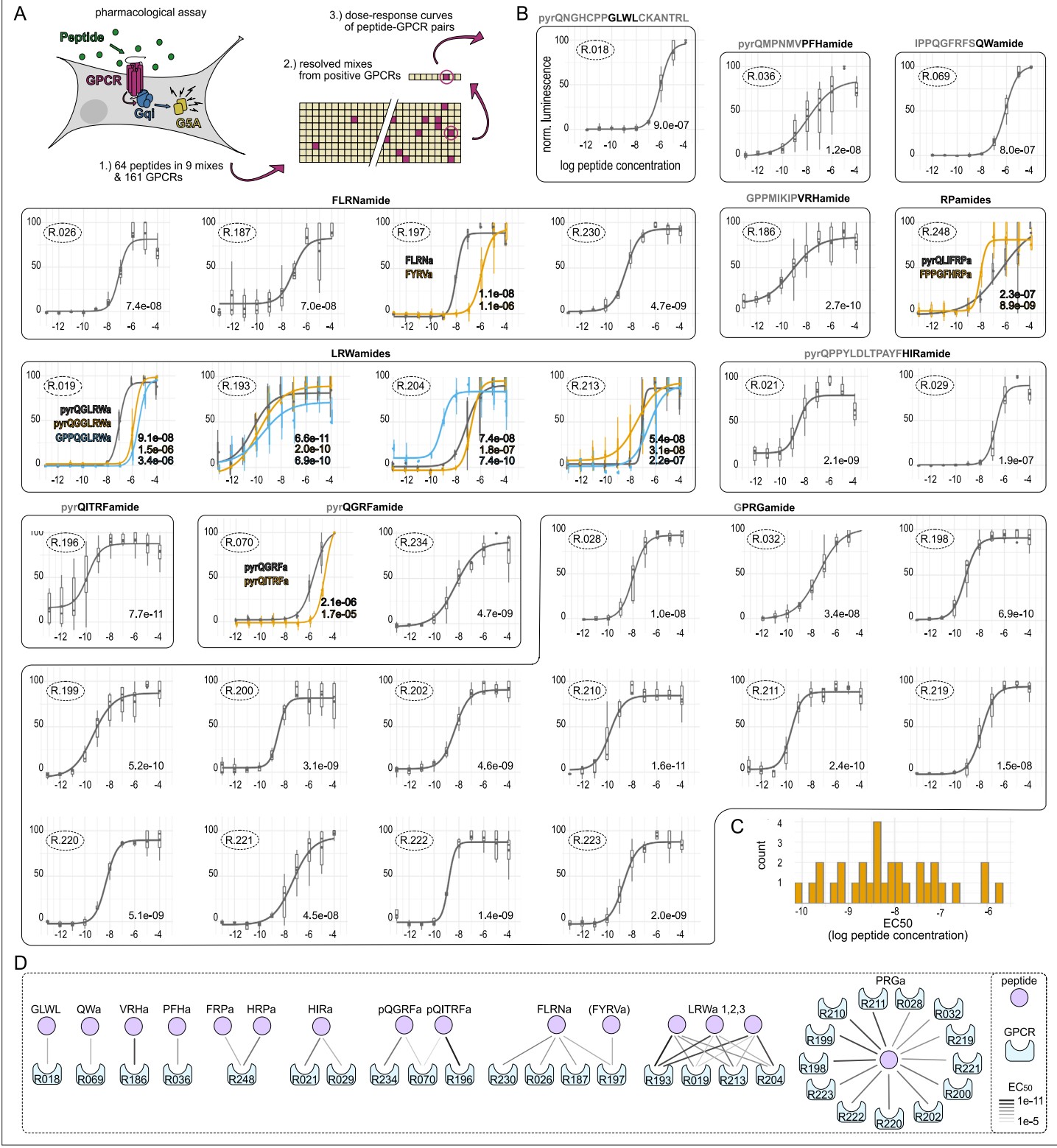

**Figure 3.** Dose-response curves of *Nematostella* neuropeptide G protein-coupled receptor (GPCR) pairs. (**A**) Pharmacological assay and pipeline to identify peptide-GPCR pairs. (**B**) Dose-response curves of peptide-GPCR pairs with log peptide concentration plotted against normalized luminescence. GPCRs that are activated by the same peptide(s) are grouped together with peptide sequence shown above and peptide name highlighted in black. If several peptides activate the same receptor, peptide sequences are shown within the graph. Receptor identification number is encircled in the upper left of each curve, $EC_{50}$ values are indicated in the lower right. Sample size per datapoint = 9. Error bars show distribution of datapoints with box

*Figure 3 continued*

indicating upper and lower quartile. (**C**) Histogram of $EC_{50}$ values of peptide-GPCR pairs, showing only the lowest $EC_{50}$ per GPCR. (**D**) Peptide-receptor pairings showing number of receptors activated by the different peptides. Connection strength indicates $EC_{50}$ values.

The online version of this article includes the following source data and figure supplement(s) for figure 3:

**Source data 1.** Tibble with all data points used to calculate the dose-response curves and $EC_{50}$ values, in .csv format.

**Figure supplement 1.** Dose-response curves with $EC_{50}$ values and peptide precursor of different HIRamide peptides.

**Figure supplement 1—source data 1.** Tibble with all data points used to calculate the dose-response curves and $EC_{50}$ values, in .csv format.

**Figure supplement 2.** Dose-response curves with $EC_{50}$ values and peptide precursor of different PRGamide peptide versions for two of the PRGamide receptors.

(GLWL/R18 $EC_{50}$=9E-7, PFHa/R36 $EC_{50}$=1.2E-8, VRHa/R186 $EC_{50}$=2.7E-10, QWa/R69 $EC_{50}$=9E-7). Peptides ending in RPamide from the HRPamide and FRPamide precursors activate the same, single receptor R248. The HRPa peptide has a lower $EC_{50}$ value, the FRPa peptide, however, has a lower threshold activation while reaching its maximum activation at higher concentrations with an overall more shallow curve slope (HRPa/R248 $EC_{50}$=8.9E-9, HRPa/R248 slope = 1.75, FRPa/R248 $EC_{50}$=2.3E-7, FRPa/R248 slope = 0.42) (***Supplementary file 10***).

The RFamide peptides with the sequence pyrQGRFamide and pyrQITRFamide are each encoded on separate precursors and despite their strong sequence similarities each activate separate receptors (pyrQGRFa/R70 $EC_{50}$=2.1E-6, pyrQGRFa/R234 $EC_{50}$=4.7E-9, pyrQITRFa/R196 $EC_{50}$=7.7E-11) although one of the two pyrQGRFamide receptors is also activated by higher concentrations of pyrQITRFamide (QITRFa/R70 $EC_{50}$=1.7E-5). Given that R70 has $EC_{50}$ values in a lower micromolar range for both RFamides, it is possible that this receptor has another unknown ligand that is more specific but shares some structural similarity to these RFamides.

Two receptors are activated by HIRamide (R21 $EC_{50}$=2.1E-9, R29 $EC_{50}$=1.9E-7). The more sensitive R21, however, showed a strong base activation at all tested concentrations, leading to a shifted $EC_{50}$ for which we adjusted the minimum values (***Figure 3—figure supplement 1***). We also tested different copy versions of HIRamide peptides and most activated the two receptors in a similar concentration range (***Figure 3—figure supplement 1***).

Four receptors are activated by FLRNamide (FLRNa/R26 $EC_{50}$=7.4E-8, FLRNa/R187 $EC_{50}$=7E-8, FLRNa/R197 $EC_{50}$=1.1E-8, FLRNa/R230 $EC_{50}$=4.7E-9), and one of these is also sensitive to higher concentrations of FYRVamide (FYRVa/R197 $EC_{50}$=1.1E-6). However, we only tested the non-modified FLRNamide peptide and not the phenylacetyl-LRNamide as described to exist in sea anemones (***Grimmelikhuijzen et al., 1990***) which may be a better ligand.

The three LRWamide peptides pyrQGLRWamide (LRWa1), pyrQGGLRWamide (LRWa2), and GPPQGLRWamide (LRWa3), which are each encoded as a single copy on separate precursors (***Figure 1C***), cross-activate four different GPCRs. Each LRWamide seems to have one preferred receptor (LRWa1/R19 $EC_{50}$=9.1E-8, LRWa2/R213 $EC_{50}$=3.1E-8, LRWa3/R204 $EC_{50}$=7.4E-10) plus a fourth GPCR (R193) which appears similarly sensitive to all LRWamide peptides (LRWa1/R193 $EC_{50}$=6.6E-11, LRWa2/R193 $EC_{50}$=2E-10, LRWa3/R193 $EC_{50}$=6.9E-10). Receptors 19 and 204 have clearly shifted curves and lower $EC_{50}$ values for their preferred peptide, while receptor 213 has similar $EC_{50}$ values for LRWa1 ($EC_{50}$=5.4E-8) and LRWa2 ($EC_{50}$=3.1E-8) but is more sensitive to LRWa2 at lower concentrations while reaching its maximum activation later.

Finally, the PRGamide peptide, which belongs to the ancestral cnidarian PRXamides, activates at least 13 different receptors (R28, R32, R198, R199, R200, R202, R210, R211, R219, R220, R221, R222, R223) in *Nematostella* with $EC_{50}$ values between 1.6E-10 and 4.5E-8. We also tested R28 and R32 with longer versions of the PRGamide, as these were described elsewhere based on mass spectrometry data (***Hayakawa et al., 2019***). The higher $EC_{50}$ values for these longer PRGamides (***Figure 3—figure supplement 2***) and the sequences on the precursor themselves (***Supplementary file 2***), however, suggest that these are not fully processed and the actual PRGamide is a tetrapeptide (GPRGamide and APRGamide) as previously suggested (***Koch and Grimmelikhuijzen, 2020***). The same is likely true for the longer QGRFamide version (QGRFGREDQGRFamide) (***Hayakawa et al., 2019***) which is also likely not fully processed as in this case the activation of the QGRFamide receptors in the initial screen was much lower for the mix that contained the longer version than for the mix containing the fully processed pyrQGRFamide peptide (***Supplementary file 4*** – mix 4 vs. mix 6).

## At least nine neuropeptide GPCR families are ancestral to cnidarians

To reconstruct the evolution of neuropeptide receptors in animals, we analyzed the phylogenetic relationships of the 31 deorphanized *N. vectensis* neuropeptide GPCRs to other cnidarian and bilaterian GPCRs. From the cluster map (*Figure 2*), we chose sequences with connection to the bilaterian and cnidarian neuropeptide GPCR clusters and calculated phylogenetic trees. In an initial analysis, we found that the bilaterian chemokine, purino, fatty acid, and other related bilaterian-specific receptors are likely a diverged ingroup of the bilaterian neuropeptide gamma rhodopsin receptors (*Figure 4—figure supplements 1 and 2*, *Supplementary file 11*). Gamma rhodopsin receptors are specific to bilaterians and include somatostatin/allatostatin A, opioid/somatostatin/allatostatin C, kisspeptin, and melanin-concentrating hormone receptors (*Mirabeau and Joly, 2013*; *Thiel et al., 2021*). We then deleted redundant group-specific expansions of loosely connected orphan clusters, the non-neuropeptide chemokine and related receptors, decreased the number of bilaterian species, and calculated a detailed neuropeptide-GPCR phylogeny (*Figure 4*, *Supplementary file 11*).

The majority of known neuropeptide GPCRs are grouped into three main bilaterian clusters and two main cnidarian clusters (*Figure 4B*). Within the two cnidarian clusters, there are at least nine neuropeptide GPCR families that are ancestral to Cnidaria, all of which are represented with clear anthozoan and medusozoan orthology groups. The deorphanized *N. vectensis* neuropeptide GPCRs belong to at least seven of these ancestral cnidarian families (*Figure 4B and C*).

One of the two main cnidarian clusters seems to have only expanded late within the Hexacorallia branch, with an especially high number of more than 100 GPCRs in *N. vectensis* (*Figure 4B and C*). The entire cluster is expanded in *N. vectensis* and *Exaiptasia pallida*, but neither expanded in *Corallium rubrum*, nor in medusozoan species. This cluster likely expanded from a single ancestral cnidarian receptor family as it only contains a single branch of medusozoan sequences. Alternatively, this cluster can be traced back to two peptidergic systems, with a loss of medusozoan sequences in one of them. This cluster contains the more promiscuous and less sensitive *N. vectensis* QGRFamide GPR70, the QWamide GPR69, and the PFHamide GPR36. The second major cnidarian neuropeptide GPCR cluster contains at least eight ancestral cnidarian receptor families, based on the presence of clear anthozoan and medusozoan orthology groups, with deorphanized *N. vectensis* receptors in six of them (*Figure 4B and C*). The QITRFamide GPR196 belongs to a small group of hexacorallian receptors with no clear orthologs in *C. rubrum* or any medusozoan species, which might either represent a separate ancestral group with no medusozoan representative sequences or a strongly diverged Hexacorallia subcluster. This is in accordance with the absence of QITRFamide peptides in Octocorallia and Medusozoa (*Koch and Grimmelikhuijzen, 2020*). The QGRFamide GPR234 belongs to an ancestral cnidarian GPCR family, which is in accordance with the existence of QGRFamide or GRFamide peptide precursors across Cnidaria (*Koch and Grimmelikhuijzen, 2020*; *Koch and Grimmelikhuijzen, 2019*). This group is slightly expanded in the medusozoans *C. hemisphaerica* and *H. vulgaris*. The entire QGRFamide family seems related to the LRWa GPR19 containing group, which has otherwise no direct orthologous sequences in medusozoans. The GLWL peptide receptor GPR18 belongs to an ancestral cnidarian family that shows a slight expansion in *N. vectensis*. The receptors for the peptides LRWamide, FLRNamide, and HIRamide are closely related as part of an anthozoan/hexacorallian expansion, curiously twice in unrelated families. Each of the families is ancestral to cnidarians and one expanded in anthozoans and additionally contains the receptors for VRHamide and RPamides, while the other group only showed several sequences in the two analyzed Hexacorallia species *N. vectensis* and *E. pallida*. Both families contain receptors for the three peptides, LRWamide, FLRNamide, and HIRamide, and both families only contain medusozoan sequences from *Rhopilema esculentum* and *Calvadosia cruxmelitensis*. The 13 PRGamide receptors of *N. vectensis* belong to two separate ancestral cnidarian families, both of which are expanded in anthozoans. One PRGamide family independently also expanded in *H. vulgaris*. The other PRGamide family contains the *Clytia* MIH receptor which is activated by *Clytia* PRXamides (*Quiroga Artigas et al., 2020*), confirming a peptide-receptor coevolution in cnidarians for the ancestral PRXamides (*Koch and Grimmelikhuijzen, 2019*) and their receptor(s).

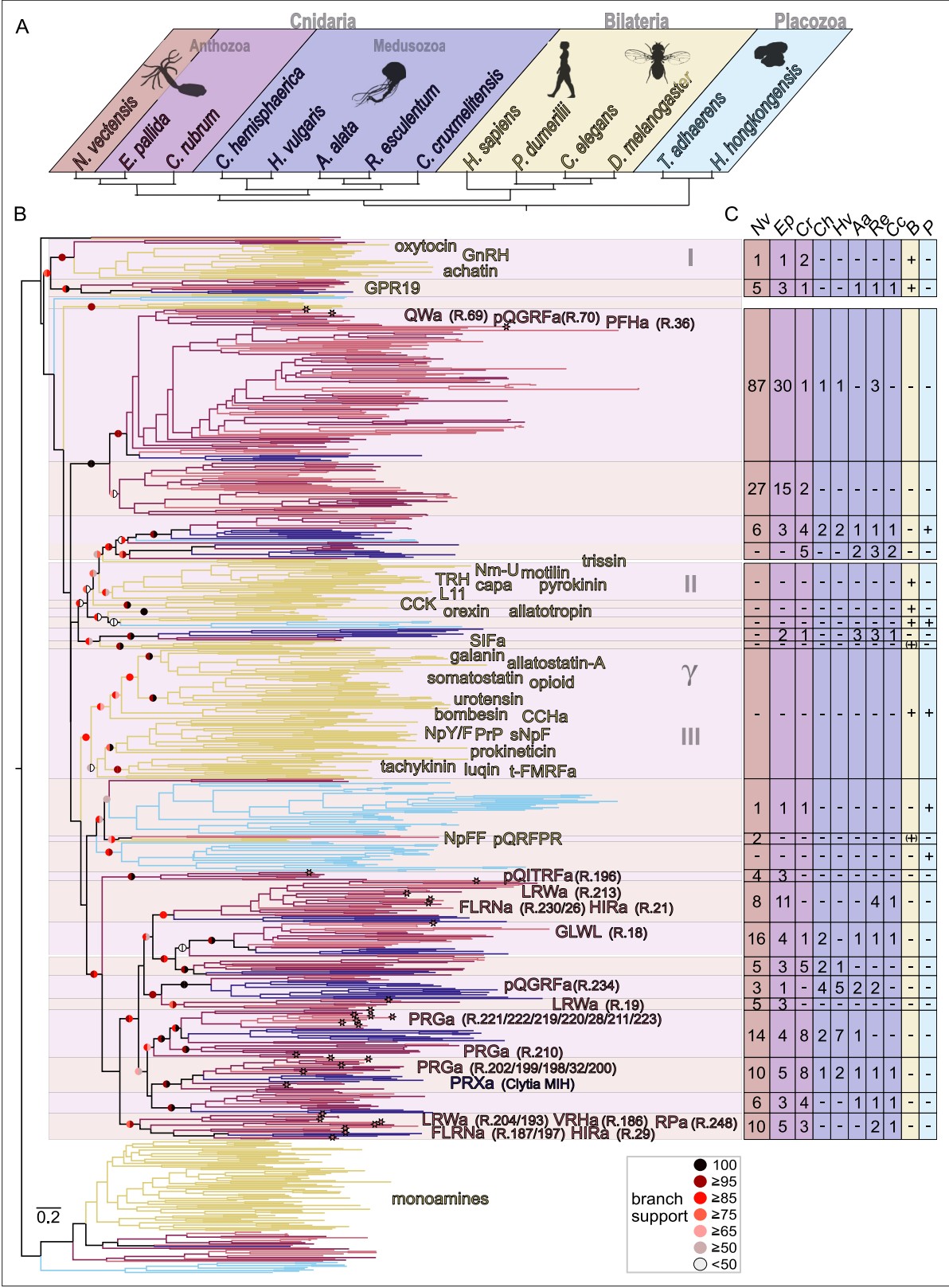

**Figure 4.** Phylogeny of metazoan class A neuropeptide G protein-coupled receptors (GPCRs). (**A**) Phylogeny of species used in B. (**B**) Phylogeny of neuropeptide GPCRs with names of ligands. Branches are color-coded according to A. Branches of deorphanized *Nematostella* GPCRs end in an asterisk. Alternating shades behind the tree branches highlight different monophyletic groups. Roman numbers 1–3 and Greek symbol gamma indicate approximate neuropeptide clusters shown in *Figure 2*. Left half circle of branch support indicates aBayes and the right half circle aLRT-SH-like support

*Figure 4 continued on next page*

*Figure 4 continued*

values. Detailed annotations in ***Supplementary file 11***. (**C**) Table with number of receptors per group as highlighted in receptor phylogeny with a straight line indicating no receptor present. Two-letter abbreviations on top correspond to species in A. Abbreviations: a=amide, B=Bilateria, CCK = cholecystokinin, GnRH = gonadotropin releasing hormone, MIH = maturation-inducing hormone, Nm-U=neuromedin U, NpFF = neuropeptide FF, NpY/F=neuropeptide Y/neuropeptide F, P=Placozoa, PrP = prolactin releasing peptide, R.#=*Nematostella* GPCR number, sNpF = short neuropeptide F, t-FMRFa=trochozoan FMRFamide, TRH = thyrotropin releasing hormone.

The online version of this article includes the following source data and figure supplement(s) for figure 4:

**Source data 1.** Raw sequences used for tree building, .fasta format.

**Source data 2.** Aligned sequences used for tree building.

**Source data 3.** Trimmed sequence alignment used for tree building.

**Source data 4.** Tree file in nexus format.

**Figure supplement 1.** Tree (FastTree) of neuropeptide G protein-coupled receptors (GPCRs) with bilaterian chemokine and related receptors.

**Figure supplement 1—source data 1.** Raw sequences used for tree building, .fasta format.

**Figure supplement 1—source data 2.** Aligned sequences used for tree building.

**Figure supplement 1—source data 3.** Trimmed sequence alignment used for tree building.

**Figure supplement 1—source data 4.** Tree file in nexus format.

**Figure supplement 2.** Tree (IQtree) of neuropeptide G protein-coupled receptors (GPCRs) with bilaterian chemokine and related receptors.

**Figure supplement 2—source data 1.** Raw sequences used for tree building, .fasta format.

**Figure supplement 2—source data 2.** Aligned sequences used for tree building.

**Figure supplement 2—source data 3.** Trimmed sequence alignment used for tree building.

**Figure supplement 2—source data 4.** Tree file in nexus format.

## Cnidarian and bilaterian neuropeptide GPCR systems expanded after the cnidarian-bilaterian split from a few ancestral systems

Our phylogenetic analyses divided the cnidarian and the bilaterian neuropeptide receptors into very few major clusters, each containing either bilaterian or cnidarian sequences (***Figure 4B***). The relationship of these major clusters to each other is not well resolved. However, the general clustering (***Figure 2C***) in combination with the phylogenetic analysis (***Figure 4B***) shows that these groups represent many-to-many bilaterian-cnidarian orthology groups. Most of the bilaterian and cnidarian class A neuropeptide GPCRs therefore diversified after the bilaterian-cnidarian split. The only family with an apparently consistent one-to-one orthology of an ancestral cnidarian and bilaterian branch of receptors is that of the bilaterian orphan receptor GPR19.

In addition, there are a few branches with unclear phylogeny or restricted taxonomic breadth. A small group of anthozoan orphan receptors without medusozoan representatives shows affinity to the expanded cluster of bilaterian oxytocin/vasopressin, GnRH, achatin, and neuropeptide S/CCAP receptors, suggesting a one-to-many orthology. This affinity is consistent in different trees (compare ***Figure 4*** and ***Figure 4—figure supplements 1 and 2***, or ***Supplementary file 11***). Other orthology groups between Cnidaria and Bilateria are less clear and the potential branches in question are not stable between the different phylogenies. The bilaterian SIFamide GPCR grouped together with an orphan cnidarian branch, but varied in its position in other analyses (compare ***Figure 4*** and ***Figure 4—figure supplements 1 and 2***, or ***Supplementary file 11***). This instability is also obvious due to the fact that the SIFamide GPCR is the protostome ortholog of the deuterostome NpFF GPCRs, which would have been expected to group together. Accordingly, the NpFF receptor was also unstable in our analyses. The NpFF and SIFamide receptor group is usually well supported, but often shows long basal branches and these two together built in previous analyses a more separated branch with unstable relationship to other bilaterian receptor groups (***Mirabeau and Joly, 2013***; ***Thiel et al., 2021***). The bilaterian QRFPR branch has a likewise unstable relationship to other bilaterian neuropeptide GPCRs in previous analyses but showed affinity to the SIFamide/NpFF GPCRs in some analyses (***Jékely, 2013***; ***Thiel et al., 2018***). Accordingly, the affinity of some *N. vectensis* sequences to the QRFPR branch is also not stable in our different analyses (compare ***Figure 4*** and ***Figure 4—figure supplements 1 and 2***; ***Supplementary file 11***). Two ancestral cnidarian branches grouped together with the *Drosophila* trissin receptor, but also this is an unstable grouping not present in our supplementary

analyses (*Figure 4—figure supplements 1 and 2*). The trissin receptor is a generally peculiar bilaterian protostome receptor as no orthologous deuterostome sequences are known (*Elphick et al., 2018*; *Mirabeau and Joly, 2013*; *Thiel et al., 2021*). The orexin/allatotropin receptors showed no direct orthology to cnidarian receptors, but appear as a sister group to a branch with orphan bilaterian and placozoan GPCRs. Together, except for the GPR19 group, we could not find cnidarian receptors that show a consistent 1:1 orthology relationship to specific bilaterian receptors such as, for example, proposed for orexin/allatotropin, somatostatin, neuropeptide Y, or tachykinin receptors (*Alzugaray et al., 2019*; *Anctil, 2009*; *Krishnan and Schiöth, 2015*). This is in accordance with other analyses that used a wider array of bilaterian neuropeptide GPCRs when comparing them to cnidarian GPCR sequences (*Quiroga Artigas et al., 2020*; *Hauser et al., 2022*; *Thiel et al., 2018*) and found rather many-to-many orthologs, if any at all.

## Cell-type-specific expression of neuropeptides and GPCRs and the peptidergic connectome of *Nematostella*

To analyze tissue-level peptidergic signaling in *N. vectensis*, we mapped the expression of neuropeptide precursors and the newly deorphanized neuropeptide GPCRs to a single-cell RNAseq dataset (*Cole et al., 2024*; *Steger et al., 2022*). The single-cell data are split into two sets, one consisting of pooled stages spanning 18 hr post-fertilization to 16-day-old primary polyp (developmental set) and the second set consisting of adult tissues only, similar to *Cole et al., 2024*; *Cole et al., 2023*. The expression of GPCRs was in many cases low and only a small percentage of cells within a given cell cluster show receptor expression. We could not detect expression of GLWL receptor R18 and HIRamide receptor R29, consistent with the generally low expression of GPCRs in animals (*Fredriksson and Schiöth, 2005*; *Regard et al., 2008*; *Soave et al., 2021*; *Sriram et al., 2019*). In contrast, neuropeptide precursors are generally highly expressed in neurons (*Smith et al., 2019*) and we could detect all of them, except for the PFHamide that could not be mapped and was only present as a partial sequence in our combined transcriptome (*Figure 1C*). Most neuropeptide precursors show restricted expression in neuroglandular cells (*Figure 5B*), with the exception of the phoenixin and GLWL precursors (*Figure 5—figure supplement 1*). The broad expression of phoenixin suggests other functions for this molecule outside neuronal signaling, e.g., in mitochondrial regulation (*Dennerlein et al., 2015*; ; *Yañez-Guerra et al., 2022*).

Individual GPCRs were often restricted to a single or very few tissue types (*Figure 5B* for tissue-type overview and *Supplementary file 12* for cell-type-specific resolution). For peptides with multiple receptors, we often found distinct patterns of receptor expression. Individual PRGamide receptors are, for example, restricted to the retractor muscle (R221, R222), the gastrodermis (R211), embryonic putative stem cells (pSC) (R210), or a combination of neuroglandular cells and adult cnidocytes (R202) (*Figure 5*). An exception is the PRGamide receptor R200, which shows expression in a wide array of tissues, including adult pSC, primary germ cells, neuronal cells, immune cells, and embryonic ectodermal cells.

The different LRWamide receptors also signal to different tissue types. The receptor R204, which is most sensitive to LRWa3, is strongly expressed in embryonic endomesodermal cells and the pharyngeal ectoderm, the receptor R19, which is most sensitive to LRWa1 is found in neuroglandular cells, the receptor R213, which showed highest sensitivity to LRWa2, is found in neuroglandular and glandular mucousal cells and the receptor R193, which is similarly sensitive to all LRWamides, is only present in glandular mucousal cells. The two receptors of the ancestral QGRFamide are also differentially expressed, with the receptor R70 only present in adult neuroglandular cells, while the highly specific R234 is additionally expressed in developmental and adult glandular mucousal cells and in some developmental retractor muscle cells.

Within the neuroglandular subset, each neuronal cell type expresses a unique combination of neuropeptide precursors and GPCRs (*Figure 5—figure supplements 2 and 3*; *Supplementary file 12*). Most peptide precursors are thereby restricted to only a few cell types. PRGamide, for example, is restricted to two types of neuroglandular cells: the larval apical organ N1.L2 cells and the N2.g1 cells, which persist in adults (*Figure 5—figure supplements 2 and 3*). This restricted expression of the PRGamide precursor is also in accordance with previously published in situ hybridizations that show a restricted expression in the larval apical organ (*Gilbert et al., 2022*). Many neuronal cell types only express a low number of different neuropeptide precursors, with few exceptions such as the

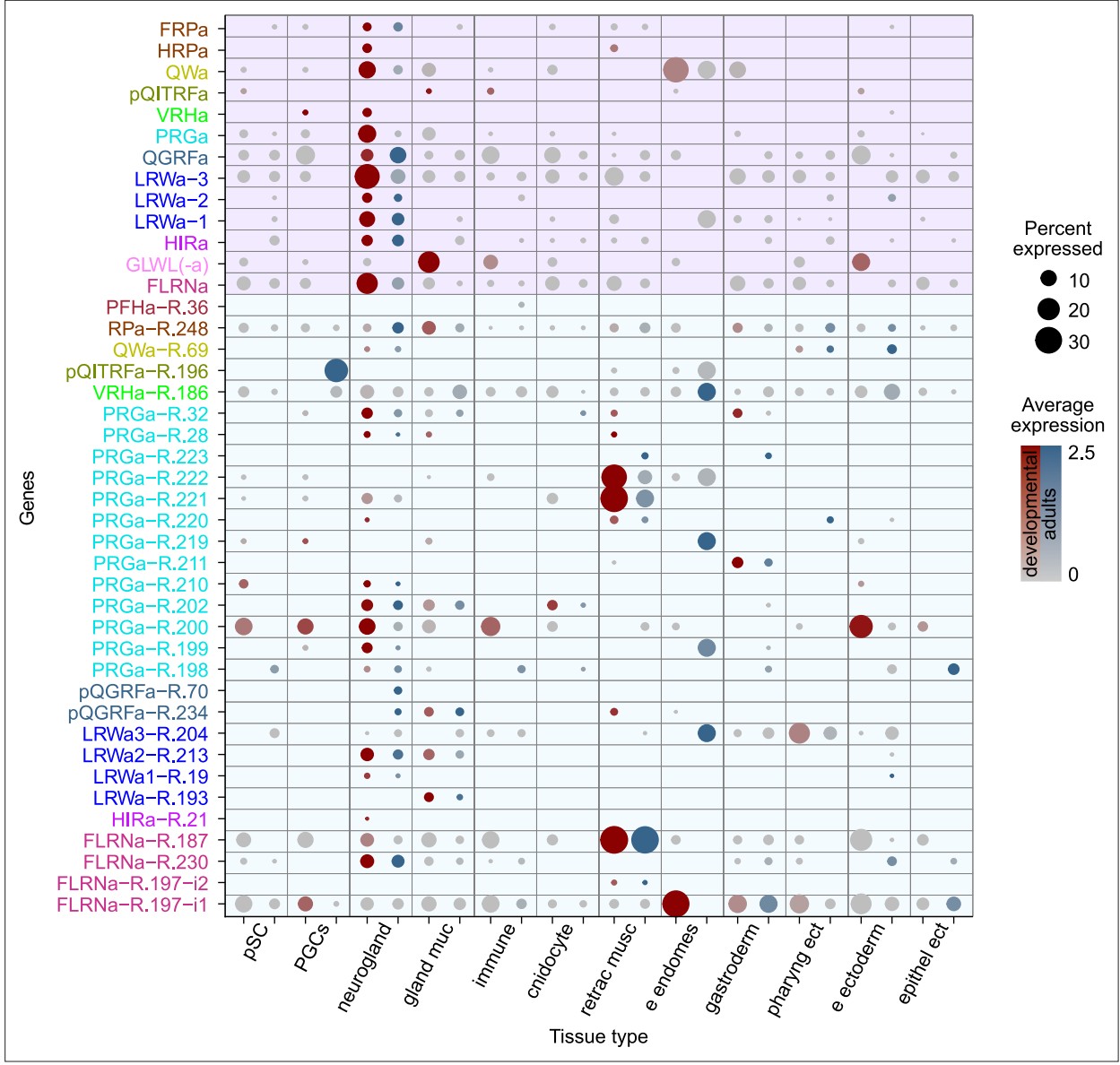

**Figure 5.** Tissue-specific expression of neuropeptide precursors and receptors (G protein-coupled receptors [GPCRs]) in *N. vectensis*. Dotplot for tissue-specific expression of peptide precursors and GPCRs. Red dots indicate expression in the developmental dataset, blue dots indicate expression in the adult dataset. Abbreviations: a=amide, e=embryonic, ect = ectoderm, endomes = endomesoderm, gland = glandular, muc = mucous, musc = muscle, neurogland = neuroglandular, PGCs = primary germ cells, pharyng = pharyngeal, pSC = putative stem cells, R=receptor (GPCR), retrac = retractor.

The online version of this article includes the following figure supplement(s) for figure 5:

**Figure supplement 1.** Tissue-specific expression of neuropeptide precursors and neuropeptide receptors (G protein-coupled receptors [GPCRs]) in *N. vectensis*.

**Figure supplement 2.** Expression of neuropeptide precursors and G protein-coupled receptors (GPCRs) in neuroglandular cell types in the developmental dataset.

**Figure supplement 3.** Expression of neuropeptide precursors and G protein-coupled receptors (GPCRs) in neuroglandular cell types in the adult dataset.

mentioned N1.L2 cells or the N1.L3 and N1.4 cells, which express between four and eight different types of neuropeptide precursors, depending on the stage. The larval N1.L2 cells express the PRGa-mide, VRHamide, QITRFamide, RPamide, LRE peptide, GLWL peptide, and the phoenixin precursor, while only expressing two receptors: the PRGamide receptor R.200 with a lower average expression and the VRHamide receptor R186. The expression of both the PRGamide and VRHamide peptide

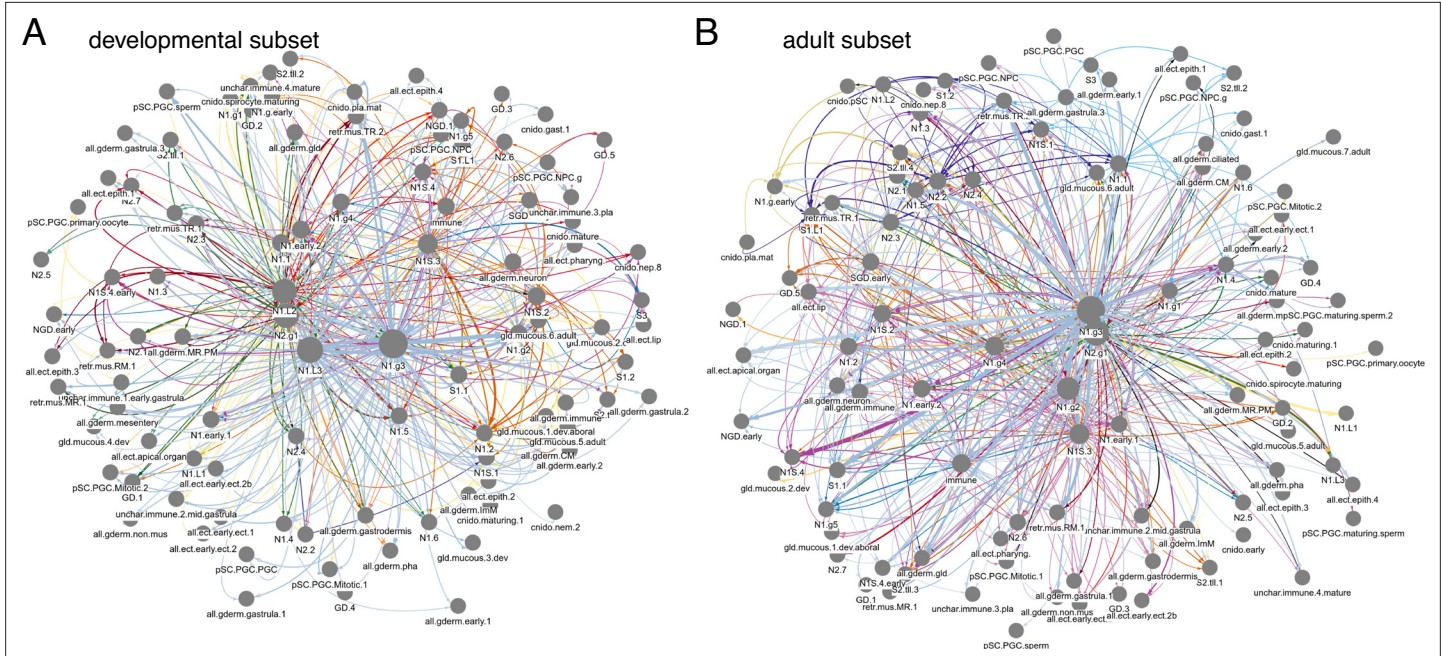

**Figure 6.** Multilayer peptidergic connectomes in *Nematostella*. Peptidergic networks in the (**A**) developmental and (**B**) adult subset. Nodes represent cell types, connections represent potential peptidergic signaling from neuropeptide-expressing cells to cells expressing one or more of the receptors for that neuropeptide. Colors represent different peptide-receptor signal channels (the different layers in the multilayer connectome).

The online version of this article includes the following source data and figure supplement(s) for figure 6:

**Source data 1.** Graph file of the multilayered peptidergic connectome in the developmental subset.

**Source data 2.** Graph file of the multilayered peptidergic connectome in the developmental subset.

**Source data 3.** Graph file of the multilayered peptidergic connectome in the adult subset.

**Source data 4.** Graph file of the multilayered peptidergic connectome in the adult subset.

**Figure supplement 1.** Multilayer peptidergic connectomes in *Nematostella*.

precursors and receptors may indicate autocrine regulation in these cells. The second PRGamide-positive cell type N2.g1 that persists in adults also shows in both datasets an expression of the PRGamide precursors and a different PRGamide receptor R.221, which is only found in few cell types and similarly may mediate autocrine feedback.

To determine the global organization of peptidergic signaling networks or the 'peptidergic connectome' (*Bentley et al., 2016*; *Deng et al., 2019*; *Smith et al., 2019*; *Williams et al., 2017*) in *Nematostella*, we constructed a multilayered network for both the developmental and the adult dataset (*Figure 6*, *Figure 6—figure supplement 1*). In these networks, nodes represent cell/tissue types and links are defined from peptide-expressing cells to receptor-expressing cells taking into account expression levels and the $EC_{50}$ values. Different peptide-receptor pairs (represented by different colors in *Figure 6*, *Figure 6—figure supplement 1B–D, F–L*) form distinct layers in this multilayer connectome. By modularity analysis, we subdivided the networks into three (developmental) or four (adult) modules, each dominated by peptides with multiple receptors: PRGamide, pyrQGRFamide, FLRNamide, and LRWamide (*Figure 6—figure supplement 1A, E*). The cells at the center of these modules that are most strongly involved in peptidergic signaling are the N1.g3, N1S.3, N1.L2, N1.L3, and N2.g1 cells in the developmental dataset and the N1.g3, N1S.3, N1.g2, and N2.g1 cells in the adult dataset.

## Discussion

We deorphanized 31 neuropeptide GPCRs of the sea anemone *N. vectensis* and reconstructed their evolution and relationship to other cnidarian orphan GPCRs and bilaterian neuropeptide GPCRs. Our phylogeny suggests that the identified neuropeptide GPCRs belong to two major

GPCR clades that expanded independently of each other. One clade diversified in stem Cnidaria into eight receptor families that are all present across medusozoans and anthozoans. The other clade diversified later within the anthozoans, likely from a single ancestral cnidarian receptor. The expansion of GPCR systems is a general feature of GPCR evolution and happened independently in different lineages and at different times during animal evolution (*Beets et al., 2022*; *Mirabeau and Joly, 2013*).

The cnidarian neuropeptide receptors we identified here show many-to-many orthology relationship to bilaterian neuropeptide GPCRs. We did not identify any direct receptor orthologs between cnidarians and bilaterians, indicating extensive parallel expansion of neuropeptide signaling in the two clades. Previously, some neuropeptides have been hypothesized to be orthologous across cnidarians and bilaterians, including neuropeptides ending in an RFamide (*Jékely, 2013*; *Walker et al., 2009*), which is a frequent motif (*Elphick and Mirabeau, 2014*; *Jékely, 2013*; *Walker et al., 2009*). We identified two *N. vectensis* RFamide receptors, one for the ancestral cnidarian neuropeptide pyrQGR-Famide (*Koch et al., 2021*; *Koch and Grimmelikhuijzen, 2019*) and one for the hexacorallia pyrQITR-Famide. These two *N. vectensis* receptors belong to either of the two major clades of cnidarian receptors, suggesting that new RFamide receptors and/or peptides can evolve independently. Likewise, within bilaterians there are many RFamides across protostomes and deuterostomes that are not orthologous to each other (*Elphick and Mirabeau, 2014*; *Thiel et al., 2021*). The cnidarian RFamide receptors are also more closely related to other cnidarian neuropeptide receptors than to bilaterian RFamide receptors, arguing against the orthology of cnidarian and bilaterian RFamides. C-terminal aromatic amino acids are found in various neuropeptides and also many monoamine receptors are activated by derivatives of aromatic amino acids (*Bauknecht and Jékely, 2017*; *Elphick and Mirabeau, 2014*). Aromatic amino acids might have been early ligands in the evolution of monoamine neurotransmission and ancestrally present as a common structural feature at the C-termini of neuropeptides activating peptide receptors.

Our phylogenetic analyses and GPCR resource will be useful to directly predict the ligand of neuropeptide receptors in other cnidarian species. We expect this because of the long-term coevolution of peptide-receptor pairs that has been extensively documented in bilaterians. The orthology of six of the identified anthozoan *N. vectensis* PRGamide receptors to the PRXamide receptor of the hydrozoan *C. hemisphaerica* is the first example of a similar long-term ligand-receptor association across cnidarians. Receptors of ancient peptides such as PRXamides or GRFamide are therefore expected to be orthologous even in distantly related cnidarians. Other Hexacorallia- or Anthozoa-specific peptides such as HIRamide, FLRNamide, and LRWamide (*Koch and Grimmelikhuijzen, 2020*) can be tested on directly orthologous receptors. Phylogenetic analyses combined with targeted deorphanization experiments of predicted orthologs could thus lead to the rapid characterization of new receptors in other cnidarians.

Our large-scale GPCR-peptide resource also allowed us to analyze tissue-level peptidergic networks in *N. vectensis*. With high-resolution single-cell resources becoming available in a larger number of species, it will be possible to predict the cellular targets of neuropeptide signals with increasing precision based on receptor expression. The unique combinations of proneuropeptides and neuropeptide receptors in the different neuronal cell types can also be used as markers to identify and characterize neuronal cell types. The uniquely specific combinatorial expression of proneuropeptides as neuronal markers parallels the situation found in bilaterian nervous systems, where proneuropeptides provide fingerprint-like identities to different neuronal cell types (*Smith et al., 2019*; *Taylor et al., 2021*; *Williams et al., 2017*).

Finally, our work also opens up new avenues in experimental neuroscience in cnidarians (*Bosch et al., 2017*). With readily available genetic manipulation techniques in *N. vectensis* and other species (*Ikmi et al., 2014*; *Nakanishi and Martindale, 2018*; *Paix et al., 2023*, *Quiroga Artigas et al., 2020*; *Sanders et al., 2018*; *Wittlieb et al., 2006*), the identified neuropeptide-receptor interactions will enable genetic manipulations of both ligand and receptor(s), to reveal the biological functions of peptidergic signaling.

Overall, we identified receptors for peptides from nearly half of the *Nematostella* neuropeptide precursors, including the receptors of the ancient PRXamide and pyrQGRFamide peptides. Future studies aiming at finding the remaining receptors could focus on GPCRs with no sequence similarity to known bilaterian neuropeptide GPCRs, including monoamine-related or leucine-rich-repeat receptors,

or other types of receptors like DEG/ENaC-related ion channels (*Gründer et al., 2022*; *Mirabeau and Joly, 2013*).

# Materials and methods

**Key resources table**

| Reagent type (species) or resource | Designation | Source or reference | Identifiers | Additional information |
|---|---|---|---|---|
| Biological sample (*N. vectensis*) | Larval, juvenile, and adult *N. vectensis* | Specimens obtained form the Marine Invertebrate Culture Unit of the University of Exeter | N/A | N/A |
| Biological sample (cDNA) | cDNA obtained from *N. vectensis* | This study | N/A | RNA extracted with Trizol and cDNA synthesized with cDNA synthesis kit according to the manufacturer's recommendation |
| Biological sample (peptide extract) | Peptide extracts obtained from *N. vectensis* | This study | N/A | Peptides extracted from *N. vectensis* according to protocol explained in Materials and methods |
| Genetic reagent (cDNA synthesis) | SuperScript III First-Strand Synthesis System | Invitrogen (from Thermo Fisher) | 18080051 | N/A |
| Genetic reagent (polymerase) | Q5 Hot Start High-Fidelity DNA Polymerase | New England Biolabs | M0493L | N/A |
| Genetic reagent (DNA assembly) | NEBuilder HiFi DNA Assembly Master Mix | New England Biolabs | E2621L | N/A |
| Genetic reagent (restriction enzyme) | EcoRV restriction enzyme | New England Biolabs | R3195L | N/A |
| Genetic reagent (restriction enzyme) | Afl2 restriction enzyme | New England Biolabs | R0520L | N/A |
| Genetic reagent (restriction enzyme) | Hind3 restriction enzyme | New England Biolabs | R3104L | N/A |
| Genetic reagent (restriction enzyme) | BamH1 restriction enzyme | New England Biolabs | R3136L | N/A |
| Genetic reagent (restriction enzyme) | EcoRI restriction enzyme | New England Biolabs | R3101L | N/A |
| Recombinant DNA reagent (plasmid) | pcDNA3.1(+) | Invitrogen (from Thermo Fisher) | V79020 | N/A |
| Recombinant DNA reagent (plasmid) | pRK5-Gqi9 | AddGene | 125711 | N/A |
| Recombinant DNA reagent (plasmid) | pcDNA3.1(+)-G5A | This study | N/A | Sequence information provided in *Supplementary file 7* |
| Recombinant DNA reagent (plasmid) | pcDNA3.1(+)-Gqi5/9 | This study | N/A | Sequence information provided in *Supplementary file 7* |
| Recombinant DNA reagent | Cloned *N. vectensis* GPCRs in pcDNA3.1(+) | Identified in this study | N/A | Full list of cloned GPCRs with sequences in *Supplementary file 8* |
| Recombinant DNA reagent | Synthesized *N. vectensis* GPCRs in pcDNA3.1(+) | Identified in this study, codon optimized and synthesized by GenScript | N/A | Full list of synthesized GPCRs with original and codon-optimized sequences in *Supplementary file 8* |
| Sequence-based reagent | Cloning primers to create Gqi5/9 | This study | N/A | Sequence information provided in *Supplementary file 7* |
| Sequence-based reagent | GPCR cloning primers | This study | N/A | Full list of primers with sequences in *Supplementary file 8* |
| Peptide, recombinant protein | Custom peptides | Identified in this study, synthesized by GenScript | N/A | Full list of peptides with sequences and purity in *Supplementary files 2 and 3* |

*Continued on next page*

*Continued*

| Reagent type (species) or resource | Designation | Source or reference | Identifiers | Additional information |
|---|---|---|---|---|
| Commercial assay or kit (PCR clean-up) | Monarch PCR and DNA Cleanup Kit (5 μg) | New England Biolabs | T1030L | N/A |
| Commercial assay or kit (Miniprep) | GeneJET Plasmid Miniprep Kit | Thermo Fisher Scientific | K0503 | N/A |
| Strain, strain background (*E. coli*, TOP10) | One Shot TOP10 Chemically Competent *E. coli* | Thermo Fisher Scientific | C404010 | *E. coli* strain used for general cloning |
| Cell line (HEK293) | HEK293 Cells expressing GFP-AEQUORIN in Cytoplasma | Angio-Proteomie | cAP-0200GFP-AEQ-Cyto | N/A |
| Chemical compound, drug | DMEM | Gibco (from Thermo Fisher) | 10566016 | N/A |
| Chemical compound, drug | OptiMEM | Gibco (from Thermo Fisher) | 11058021 | N/A |
| Chemical compound, drug | FBS | Gibco (from Thermo Fisher) | 10500064 | N/A |
| Chemical compound, drug | PEI (polyethylenimine, 25k Mw) | Sigma-Aldrich | 408727 | Used 0.3 μl of a 1 mg/ml stock solution per 100 ng DNA |
| Chemical compound, drug | Transfectamine 5000 | AAT Bioquest | 60022 | Used 0.3 μl per 100 ng of DNA |
| Chemical compound, drug | Coelenterazine-H | Promega | S2011 | Diluted to 2 mM in ethanol and used at a final concentration of 4 μM in the assays |
| Chemical compound, drug | TRIzol Reagent | Invitrogen (from Thermo Fisher) | 15596026 | N/A |
| Other | Corning 96 Well White Polystyrene Microplate | Corning (from Sigma-Aldrich) | CLS3903 | Cell culture-treated assay plates |
| Other | Nunc EasYFlask T75 Cell Culture Flasks | Nunc (from Thermo Fisher) | 156499 | Cell culture flasks |
| Other | Flexstation 3 Multimode Microplate Reader | Molecular Devices | N/A | Plate reader |
| Software, algorithm | SoftMax Pro 7 | Molecular Devices | N/A | N/A |
| Software, algorithm | R | https://cran.rstudio.com/ | N/A | N/A |
| Software, algorithm | RStudio | https://posit.co/download/rstudio-desktop/ | N/A | N/A |
| Software, algorithm | SignalP4.1 | https://services.healthtech.dtu.dk/services/SignalP-6.0/ | N/A | N/A |
| Software, algorithm | NeuroPID | https://bio.tools/neuropid | N/A | N/A |
| Software, algorithm | HMMER3.1b2 | http://hmmer.org/download.html | N/A | N/A |
| Software, algorithm | CD-HIT | https://sites.google.com/view/cd-hit | N/A | N/A |
| Software, algorithm | TransDecoder v5.5.0 | https://github.com/TransDecoder/TransDecoder, **Haas, 2024** | N/A | N/A |
| Software, algorithm | CLANS (desktop version) | https://mybiosoftware.com/clans-20101007-visualize-protein-families-based-pairwise-similarity.html | N/A | N/A |
| Software, algorithm | CLANS (online toolkit) | https://toolkit.tuebingen.mpg.de/tools/clans | N/A | N/A |

*Continued on next page*

*Continued*

| Reagent type (species) or resource | Designation | Source or reference | Identifiers | Additional information |
|---|---|---|---|---|
| Software, algorithm | Phobius | https://phobius.sbc.su.se/data.html | N/A | N/A |
| Software, algorithm | Muscle alignment tool | https://drive5.com/muscle5/ | N/A | N/A |
| Software, algorithm | MAFFT v7 | https://mafft.cbrc.jp/alignment/software/ | N/A | N/A |
| Software, algorithm | trimAl | http://trimal.cgenomics.org/trimal | N/A | N/A |
| Software, algorithm | IQ-tree2 | http://www.iqtree.org/ | N/A | N/A |
| Software, algorithm | Fasttree | http://www.microbesonline.org/fasttree/ | N/A | N/A |

## Transcriptomic resources

We collected transcriptomes and protein predictions from different metazoans (Cnidaria: *N. vectensis*, *Alatina alata, C. cruxmelitensis, C. hemisphaerica, C. rubrum, E. pallida, H. vulgaris, Polypodium hydriforme, R. esculentum*. Bilateria: *D. melanogaster, C. elegans, Homo sapiens, P. marinus, P. dumerilii, Saccoglossus kowalevskii*. Placozoa: *Hoilunga hongkongensis, Trichoplax adhaerens*. Porifera: *Amphimedon queenslandica, Ephydatia muelleri, Oscarella carmella, Sycon ciliatum, Tethya wilhelma*, Ctenophora: *Pleurobrachia bachei, Mnemiopsis leydi, Hormiphora californiensis*) and the filasterian *Tunicaraptor unikontum*. Transcriptomic databases were translated to protein sequences using the tool TransDecoder v.5.5.0 (*Haas, 2024*, http://transdecoder.github.io/) with a minimum length of 60 amino acids. For completeness assessment of the transcriptomes, we ran BUSCO v5.2.1 in protein mode and with the lineage database 'eukaryota_odb10' (database creation: September 2022; number of BUSCOs: 255). The source of the databases used for this analysis and the results of the completeness analysis are available in *Supplementary file 5*. Different transcriptomes of *N. vectensis* (http://metazoa.ensembl.org/species.html, https://hdl.handle.net/1912/5613, https://simrbase.stowers.org/starletseaanemone) were translated into protein sequences and merged, followed by the use of CD-hit (*Fu et al., 2012*; *Li and Godzik, 2006*) with a similarity setting of 0.85.

## Neuropeptide precursor search

Neuropeptide precursors of *N. vectensis* were identified with different bioinformatic strategies. First, we carried out BlastP analyses based on previously published datasets (*Hayakawa et al., 2019*; *Koch and Grimmelikhuijzen, 2020*; *Thiel et al., 2021*; *Elkhatib et al., 2022*). Sequences with e-values <1E-02 were manually scanned for the presence of multiple cleavage sites and similarity to known proneuropeptides. Additionally, we obtained a predicted secretome by using SignalP4.1 with the sensitive option (D-cutoff 0.34). This secretome was then used to search for novel precursors by two different methodologies. Pattern searches were done as described before (*Thiel et al., 2021*), based on repetitive cleavage sites. The resulting sequences were then manually checked for occurrence of similar motifs between these cleavage sites. The obtained secretome was also scanned with the machine-learning algorithm NeuroPID that enriched the number of single-copy neuropeptide precursors (*Ofer and Linial, 2014*). This last methodology, however, produced a large database that included thousands of hits, with a high level of false positive proteins that contain a signal peptide and any number of monobasic or dibasic sites. The list obtained with NeuroPID was then used as a separate database for our mass spectrometry analysis to confirm hits without repetitive motifs. A list of proneuropeptides is provided in *Supplementary file 2*.

## Peptidomics and mass spectrometry analysis

*N. vectensis* specimens were obtained from a culture maintained at the Marine Invertebrate Culture Unit at the University of Exeter. We processed four samples for peptidomics. Two samples contained larvae of different ages and primary polyps up to the age of 10 days. The other two samples contained juveniles and adult tissue. Feeding-stage animals were starved for 2 days prior to collection. All samples were quickly rinsed with Milli-Q water and snap-frozen in liquid nitrogen. Each sample was

manually homogenized with a mortar and pestle in 10 ml of ice-cold acidified methanol (90% methanol, 9% water, 1% acetic acid). The homogenate was collected and sonicated on dry ice for 4×15 s with a 30 s rest between cycles. Samples were centrifuged (10 min × 4000×*g*) and the supernatant transferred to a new tube. The supernatant was concentrated in a vacuum concentrator until all methanol was evaporated. Samples were again centrifuged (10 min × 16,000×*g*) and the supernatant was twice delipidated, each time using 2 ml n-hexane and recovering the aqueous layer. Samples were then desalted with Pierce C18 spin columns following the manufacturer's guidelines and dried in a vacuum concentrator. Prior to LC-MS/MS analysis, samples were re-suspended in 5% acetonitrile, 95% water, 0.1% formic acid.

Samples were analyzed on a Waters nanoACQUITY UPLC coupled to a QExactive mass spectrometer (Thermo Scientific, Bremen, Germany) equipped with a nano-electrospray ion source. The column (µPAC trapping column, Thermo Scientific) was loaded with 5 µl of sample and set to a flow rate of 750 nl per minute. A linear gradient of solvent B (98% acetonitrile, 0.1% formic acid, 2% HPLC grade water) starting at 1% and increasing to 40% in solvent A (2% acetonitrile, 0.1% formic acid, 98% HPLC-grade water) over 80 min was used to separate peptides. MS data were acquired in a Top20 data-dependent acquisition mode with a dynamic exclusion of 20 s. The most abundant precursor ions from a full-scan MS were selected for higher-energy collisional dissociation fragmentation.

Full MS1 scans were acquired with a resolution of 70,000 with automatic gain control (AGC) set to 3E+6, a maximum injection time of 100 ms and a scan range of 350–1850 m/z. The MS/MS fragmentation scans had a resolution of 17,500, AGC of 1E+5, a maximum injection time of 80 ms, and a normalized collision energy of 28.

Raw LC-MS/MS data were analyzed using PEAKS Studio X+ (v10.5 Bioinformatics Solutions Inc, Canada). Peptides were identified using a database of sequences generated from the entire secretome, the NeuroPID predictions, and the predicted precursors from BLAST and motif searches. The precursor mass error tolerance was set to 5 ppm and the fragment mass error tolerance to 0.02 Da. The following variable post-translational modifications were included in the database search: pyro-glutamation of N-terminal glutamic acid (–18.01 Da) or glutamine (–17.03 Da), C-terminal amidation (–0.98 Da) and half of a disulfide bridge on cysteine (–1.01 Da). Oxidation of methionine (+15.99 Da) was also included as a variable modification. Since the samples are not enzymatically digested, the database search parameters included 'No Enzyme' and digest mode of 'Unspecific'.

A false discovery rate of <1% was applied to MS/MS peptide identifications and the resulting list of peptides was exported. Selection of candidate neuropeptides were based on the presence of a signal sequence in the precursor and of peptides being flanked or containing different potential cleavage site motifs. N-terminal cleavage site motifs included: KR/RR/RK/KK/EE/DD/ED/DE directly flanking the N-terminus of the peptide or Q, xP, or xxP as the most N-terminal amino acid of the peptide (with x standing for a variable amino acid). C-terminal cleavage site motifs included KK/KR/RR/RK flanking the C-terminus of the peptide or GR/GK flanking the C-terminus of amidated peptides, with G denoting the donor of the amide group. The resulting list contained neuropeptide candidates where the MS/MS spectra were manually inspected to verify quality and confidence. To prioritize peptides for further investigation, each identification was categorized as 'confident', 'uncertain', or 'poor'. 'Confident' identifications were characterized by MS/MS spectra where peaks could be clearly distinguished from noise, the b and y ion ladders resolved amino acid masses with excellent peptide coverage missing at most one or two ions. 'Uncertain' peptide identifications were characterized by having either (1) ion ladders with gaps of three amino acids but maintaining good overall coverage when the entire peptide was considered or (2) the identified sequence overlapped with the predicted signal peptide or (3) disulfide bridges were present resulting in unresolved fragmentation between the cysteine-cysteine bond, or (4) fragmentation was consistent with a confident peptide however the sequence was thought to be intermediate peptide requiring further biological processing. If MS/MS spectra had ion ladders with gaps larger than 3 amino acids or low intensity peaks, the identification was considered 'poor'.

All mass spectrometric data are available through the PRIDE repository (*Perez-Riverol et al., 2022*) with accession number PXD041235. Peptides and details about detection, flanking amino acids, precursors sequences categorized into 'confident', 'uncertain', and 'poor' are provided in *Supplementary file 1*.

## GPCR sequence analysis

To identify potential neuropeptide receptors from the GPCR family A (the most extensive neuropeptide family), the full sequence alignment of the class A GPCRs (PF00001) was obtained from the PFAM database (https://pfam.xfam.org). The alignment was used to produce a hidden Markov model (HMM) with hmmer-3.1b2 (*Eddy, 2011*), which was then used to mine the proteomes from the species described above with an e-value cutoff of 1E-10. Redundant sequences were removed using CD-Hit (*Eddy, 2011*; *Fu et al., 2012*) with a similarity setting of 0.95. All GPCR protein sequences are provided in *Supplementary file 6*. The obtained sequences were analyzed using Phobius (*Käll et al., 2007*) to predict the number of transmembrane domains and only sequences with a minimum of four and maximum of nine transmembrane domains were kept for further analyses. The relationship between the obtained proteins from the different species was analyzed using an all-vs-all BLAST-based cluster strategy with the CLANS software (*Eddy, 2011*; *Frickey and Lupas, 2004*; *Fu et al., 2012*). The initial all-vs-all BLAST file was created using the online CLANS toolkit (https://toolkit.tuebingen.mpg.de/tools/clans), with the default BLOSUM62 scoring matrix and BLAST HSPs extracted up to e-values of 1E-14. The sequences were then clustered using the CLANS desktop version with a p-value cutoff of 1E-25 and color-coded according to taxonomy. Experimentally confirmed annotated sequences from human, *D. melanogaster*, *C. elegans,* and *P. dumerilii* were used as reference sequences to annotate the cluster maps. Original cluster-map files including all sequences are provided in *Figure 2—source data 1* and *Figure 2—figure supplement 1—source data 1*. *N. vectensis* sequences with connection to the bilaterian neuropeptide GPCR cluster or belonging to orphan clusters ancestral to cnidarians were cloned for experimental testing.

For the phylogenetic analysis, we extracted the sequences of the bilaterian and cnidarian neuropeptide GPCR cluster and those connected to it. As an outgroup, we chose a subcluster of monoamine receptors that showed the strongest connection to the main neuropeptide GPCR cluster. Initial analyses were done by aligning sequences with muscle (*Edgar, 2004*), trimming the alignment with the gappyout function of TrimAl (*Capella-Gutiérrez et al., 2009*) and calculating trees with FastTree (*Price et al., 2009*) using the lg model. In subsequent analyses, we aligned the extracted genes with MAFFT v7 using the iterative refinement method E-INS-i (*Katoh et al., 2002*). Alignments were trimmed with TrimAl in gappyout mode and maximum likelihood trees calculated with IQ-tree2 with the LG + G4 model (*Minh et al., 2020*). Branch support was calculated by running 1000 replicates with the aLRT-SH-like and aBayes methods. Protein sequences, untrimmed and trimmed alignments, and tree files are provided in *Figure 4—source data 1–4*, *Figure 4—figure supplement 2—source data 1–4*. The detailed trees shown in *Figure 4* and *Figure 4—figure supplements 1–2* with annotated branches are provided in *Supplementary file 11*.

## GPCR and reporter gene cloning

All GPCRs were cloned from cDNA into the pcDNA3.1(+) vector either by standard cloning strategies based on restriction enzymes or by using the NEBuilder HiFi DNA Assembly kit. Genes that proved more problematic to clone were codon optimized and synthesized into a pcDNA3.1(+) vector by Genscript synthesis services. All tested GPCR sequences with individual cloning strategy, primers used, and codon-optimized sequences are provided in *Supplementary file 8*. The chimeric Gqi9 protein was ordered from Addgene (Plasmid No. 125711). This was then further modified using PCR and cloned into the pcDNA3.1(+) vector using the NEBuilder HiFi DNA Assembly kit. The reverse primer was engineered to change the most C-terminal amino acid residues Tyr-Cys-Gly-Leu-Cys to Asp-Cys-Gly-Leu-Phe, making it similar to the promiscuous chimeric Gqi5 protein (*Conklin et al., 1993*) yielding the vector pcDNA3.1-Gqi5/9. The chimeric G5A GFP-Aequorin protein (*Baubet et al., 2000*) was codon optimized for human cells and synthesized into a pcDNA 3.1(+) vector by Genscript. Codon-optimized G5A and Gqi5/9 sequences as well as cloning primers and further details are provided in *Supplementary file 7*.

## Cell transfection and deorphanization assay

A detailed step-by-step protocol has been published in *Thiel et al., 2023*. In brief, for transfection we used HEK293 cells that stably express the chimeric GFP-Aequorin protein G5A (Cat No. cAP-0200GFP-AEQ-Cyto). Cells were grown in 5% $CO_2$ atmosphere in DMEM (containing 4.5 g/l glucose, L-glutamine, sodium pyruvate, Thermo; Cat. No. 10566016) supplemented with 10% FBS

(heat inactivated, Thermo; Cat. No. 10082147). A confluent T75 Flask of cells was transferred into three to four clear-bottom 96-well plates and grown for 2 days. At about 90% confluency, cells were transfected either with Transfectamine 5000 (T5000) or 25 kDA branched PEI (1 mg/ml), according to the protocol from *Durocher et al., 2002*. The cell medium in the 96-well plates was exchanged with 90 µl of OptiMEM (supplemented with 5% FBS) prior to transfection. For each well, 10 µl OptiMEM (without FBS), 70 ng of GPCR containing plasmid, 70 ng of Gqi5/9 plasmid, 10–20 ng of G5A plasmid (to increase luminescence values of our HEK293 cell line), and 0.45–0.48 µl T5000/PEI were mixed and incubated for 20 min at room temperature. Transfection mixture was then added to the cells. Two days post transfection, the medium was removed and substituted with OptiMEM media supplemented with 4 µM coelenterazine-H (Promega; Cat. No. S2001), and incubated for a period of 2–3 hr. Readings were performed with a FlexStation 3 Multi-mode Microplate reader (Molecular Devices), for a period of 60 s per well, ligand injection after 15–18 s, and the whole plate was read with the Flex option. We first tested different peptide mixes on each individual GPCR with a concentration of 10 µM per peptide. Receptors that were activated by any of these mixes were then tested with the individual peptides of the activating mix at a concentration of 10 µM. Individual peptides that activated a GPCR were then tested at different concentrations between 1E-13 M and 1E-4 M to obtain dose-response curves. Each peptide-receptor pair was tested in three independent triplicates. *Supplementary file 3* contains the peptide mixes and *Supplementary file 4* the test results of the peptide mix screening assays. The readout data for the final dose-response curves is provided in *Supplementary file 9* and as .csv files in the repository in the data folder. The data were analyzed in R with the drc package for curve fitting and $EC_{50}$ calculations (*Ritz et al., 2015*). The scripts and the data in .csv format are provided on Zenodo (*Thiel et al., 2024*). The cells were tested for mycoplasma contamination by PCR.

## Single-cell analysis

Gene models corresponding to all receptor-peptide pairs are not available in the version 1 genome and accompanying gene model set (https://figshare.com/articles/dataset/Nematostella_vectensis_transcriptome_and_gene_models_v2_0/807696), however all relevant gene models were identified within the Nv2 set of gene models that accompany the vs.2 chromosome-level genome build (https://doi.org/10.1101/2020.10.30.359448). A single-cell atlas dataset for *Nematostella*, mapped to this genome and Nv2 gene model set with corresponding clustering annotations, is available (http://cells.ucsc.edu/?ds=sea-anemone-atlas; under an Nv2 subdirectory). Expression data for this gene set was extracted from this dataset, together with the cell clustering information. Expression profiles were visualized using the Seurat::DotPlot function for both the coarse tissue-level clustering and cell-type clustering of the neuroglandular complement.

## Network analysis

The multilayered peptidergic connectome was reconstructed based on the scRNAseq data from the developmental and adult subsets. We constructed an interaction network based on cell-specific average expression values of proneuropeptides and their receptors. Each cell type was a separate node in the network and connections were defined between a peptide-expressing and receptor-expressing cell based on the geometric mean of peptide and receptor expression, weighted by the absolute value of $\log_{10}EC_{50}$. We used the formula:

sqrt(PeptideExpr * ReceptorExpr) * |log(EC50)|.

Modules were identified with the Leiden algorithm (*Traag et al., 2019*). The network was visualized with the visNetwork package. The analysis was done with the script Figure6_and_Figure6_fig_suppl1.R (*Thiel et al., 2024*).

## Acknowledgements

We would like to thank Dr Cameron Hird for help with animal husbandry. We also thank Dr Francesca Carlisle for testing our cells for mycoplasma contamination. This work was funded by the Wellcome Trust (214337/Z/18/Z), the Leverhulme Trust (Project Grant RPG-2018–392), a BBSRC Discovery fellowship to LAYG (BB/W010305/1), and the KU Leuven grant C16/19/003.

## Additional information

### Competing interests

Gáspár Jékely: Reviewing editor, *eLife*. The other authors declare that no competing interests exist.

### Funding

| Funder | Grant reference number | Author |
|---|---|---|
| Leverhulme Trust | RPG-2018-392 | Daniel Thiel<br>Luis Alfonso Yañez Guerra<br>Gáspár Jékely |
| Wellcome Trust | 10.35802/214337 | Gáspár Jékely |
| Biotechnology and Biological Sciences Research Council | BB/W010305/1 | Luis Alfonso Yañez Guerra |
| KU Leuven | C16/19/003 | Amanda Kieswetter<br>Liesbet Temmerman |

The funders had no role in study design, data collection and interpretation, or the decision to submit the work for publication. For the purpose of Open Access, the authors have applied a CC BY public copyright license to any Author Accepted Manuscript version arising from this submission.

### Author contributions

Daniel Thiel, Luis Alfonso Yañez Guerra, Conceptualization, Resources, Data curation, Software, Formal analysis, Validation, Investigation, Visualization, Methodology, Writing – original draft, Project administration, Writing – review and editing; Amanda Kieswetter, Resources, Data curation, Formal analysis, Investigation, Methodology, Writing – review and editing; Alison G Cole, Resources, Data curation, Formal analysis, Investigation, Writing – review and editing; Liesbet Temmerman, Data curation, Supervision, Investigation, Methodology, Writing – review and editing; Ulrich Technau, Conceptualization, Funding acquisition, Validation, Project administration, Writing – review and editing; Gáspár Jékely, Conceptualization, Data curation, Software, Formal analysis, Supervision, Funding acquisition, Investigation, Visualization, Methodology, Project administration, Writing – review and editing

### Author ORCIDs

Daniel Thiel  http://orcid.org/0000-0003-1398-3512
Alison G Cole  http://orcid.org/0000-0002-7515-7489
Gáspár Jékely  http://orcid.org/0000-0001-8496-9836

Joint public review: https://doi.org/10.7554/eLife.90674.3.sa1
Author response https://doi.org/10.7554/eLife.90674.3.sa2

## Additional files

### Supplementary files

• Supplementary file 1. Mass spectrometry identification of *N. vectensis* neuropeptides. The table contains the results of the mass spectrometry analysis including the name, sequence and identifier of the precursor sequences, N- and C-terminal flanking sequences, and peptide sequences.

• Supplementary file 2. *N. vectensis* proneuropeptides. The file contains a list of all *N. vectensis* proneuropeptide sequences identified in this study. Each precursor is provided with a list of alternative headers from the different resources, other studies that identified the same precursor, a list of peptides that were tested in this study, the receptors that were activated and cleavage site and signal peptides are indicated by different colors.

• Supplementary file 3. Peptide mixes used for G protein-coupled receptor (GPCR) screening. The file contains the peptide mixes that were used in the initial peptide mix-GPCR screening (with additional information about the peptides and dilution of stock solutions).

• Supplementary file 4. G protein-coupled receptors (GPCR) screening with peptide mixes. The file

contains the data from the peptide mix-GPCR screening with the mixes (normalized as well as the original readings) and the readings to resolve those peptide mix-GPCR pairs that showed a positive signal and lead to the identification of the individual peptide-receptor pairs described in this study.

• Supplementary file 5. Transcriptome resources. The file contains a list of the transcriptomes used in this study, together with the sources and data from the BUSCO analysis including completeness of the transcriptomes with percentage of single, double, fragmented, and missing genes.

• Supplementary file 6. Class A G protein-coupled receptor (GPCR) sequences. The file contains all GPCR sequences that were identified from the different taxa in the HMMer screening. In .fasta format.

• Supplementary file 7. Gqi5/9 and G5A sequence. The file contains the nucleotide sequences of the Gqi5/9 and G5A gene with primer sequences and information about the mutation of the original Gqi9 that we obtained and subcloned into the pcDNA3.1(+) vector.

• Supplementary file 8. Tested *N. vectensis* G protein-coupled receptor (GPCR) sequences. The file contains a list of all tested *N. vectensis* GPCRs, including original nucleotide and protein sequences, information about the cloning strategy and either the primer sequences of genes that we cloned ourselves or the codon-optimized sequences of genes that were ordered as synthetic constructs.

• Supplementary file 9. Dose-response assay readings. The file contains the original readings from our dose-response curves as well as additional measurements such as those for supplementary figures or of peptide-receptor pairs that initially gave a signal in our screen but did not produce confident dose-response curves, as well as negative controls from peptides that were tested at different concentrations on cells that were only transfected with G5A and Gqi5/9 containing plasmids.

• Supplementary file 10. $EC_{50}$ values and curve slopes. The file contains the $EC_{50}$ values and slopes of the dose-response curves for each peptide-receptor pair.

• Supplementary file 11. Trees of sequences related to class A neuropeptide G protein-coupled receptors (GPCRs). The file contains the detailed trees that are shown in *Figure 4* and *Figure 4— figure supplements 1 and 2*, with exact support values and sequence identifiers of all branches.

• Supplementary file 12. Single-cell analysis of *N. vectensis* neuropeptide precursors and G protein-coupled receptors (GPCRs). The file contains the detailed dotplots of *Figure 5* with cell-type resolution of the developmental and the adult single-cell dataset.

• MDAR checklist

### Data availability

Figure 2—figure supplement 1—source data 1, Figure 3—figure supplement 1—source data 1, Figure 4—figure supplement 1—source data 1–4, and Figure 4—figure supplement 2—source data 1–4 contain the numerical data used to generate the figures. All mass spectrometric data are available through the PRIDE repository with accession number PXD041235. The scripts and data used for the dose-response curves, single -cell analysis, CLANS analysis, phylogenetic trees, pattern searches, and network analysis are available at Zenodo (https://doi.org/10.5281/ZENODO.10680381, *Thiel et al., 2024*).

The following dataset was generated:

| Author(s) | Year | Dataset title | Dataset URL | Database and Identifier |
|---|---|---|---|---|
| Thiel D, Kieswetter A, Temmerman L | 2024 | Deorphanization of neuropeptide GPCRs in *Nematostella vectensis* | https://www.ebi.ac.uk/pride/archive/projects/PXD041235 | PRIDE, PXD041235 |

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
