## [Editor Report · eLife assessment]

This work identifies cnidarian neuropeptides and pairs them to their GPCR, then shows that neuropeptide signaling systems have evolved and diversified independently in cnidarians and bilaterians. Neuropeptide-receptor partners were experimentally identified using established and widely used methodologies including single cell mapping, providing **compelling** evidence for the conclusions of the paper. This impressive accomplishment provides **fundamental** new insights into the evolution of neuropeptide signaling systems and will be of broad interest to neurobiologists and evolution of development researchers.

---

## [Referee Report · Joint public review]

Neuropeptide signaling is an important component of nervous systems, where neuropeptides typically act via G-protein coupled receptors (GPCRs) to regulate many physiological and behavioral processes. Neuropeptides and their cognate GPCRs have been extensively characterized in bilaterian animals, revealing that a core set of neuropeptide signaling systems originated in common ancestors of extant Bilateria. Neuropeptides have also been identified in cnidarians, which are a sister group to the Bilateria. However, the GPCRs that mediate the effects of neuropeptides in cnidarians have not been identified.

In this paper the authors perform a phylogenetic analysis of GPCRs in metazoans and report that the orthologs of bilaterian neuropeptide receptors are not found in cnidarians. This indicates that neuropeptide signaling systems have largely evolved independently in cnidarians and bilaterians. To accomplish this, they generated a library of putative and known neuropeptides computationally identified in the genome of the cnidarian sea anemone *Nematostella vectensis*. These peptides were systematically screened for their ability to activate any of the 161 putative Nematostella GPCRs.

This work identified 31 validated GPCRs. These, together with GPCRs that cluster with them, were then used to demonstrate the independent expansion of GPCRs in cnidarian and bilaterian lineages. The authors then mapped validated receptors and ligands to the Nematostella single cell data to provide an overview of the cell types expressing these signaling genes. In addition, the authors have begun to analyze neuropeptide signaling networks in *N. vectensis* by showing potential signaling connections between cell types expressing neuropeptides and cell types expressing cognate receptors.

This work is the most extensive pharmacological characterization of neuropeptide GPCRs in a cnidarian to date and thus represents an important accomplishment, and is one that will improve our understanding of how peptidergic signaling evolved in animals and its impact on evolution of nervous systems. In addition, this impressive work transforms our knowledge of neuropeptide signaling systems in cnidarians and provides the foundations for extensive functional characterization neuropeptide systems in the context of nervous systems that exhibit radial symmetry, contrasting with the bilaterally symmetrical architecture of the majority of bilaterian nervous systems.

The reviewers did not detect any weaknesses in the work but asked that the authors comment on the following points, which they have done in the revised version.

(1) Clearly, other neuropeptide signaling systems in cnidarians remain to be discovered but this paper represents a huge step forward.

(2) There are limitations in what can be interpreted from single cell transcriptomic data but the data nevertheless provide the foundations for future studies involving (i). detailed anatomical analysis of neuropeptide and neuropeptide receptor expression in *N. vectensis* using mRNA in situ hybridization and/or immunohistochemical methods and (ii). functional analysis of the physiological/behavioral roles of neuropeptide signaling systems in *N. vectensis*.

---

## [Author Response]

The following is the authors’ response to the original reviews.

Reply to comments:

(1) It was not clear why the phylogenetic analysis included non-validated GPCRs that clustered with the validated peptidergic receptors. Would restricting the phylogenetic analyses only to confirmed peptidergic GPCRs alter the topology of the tree and subsequent conclusions of independent expansion?

Thank you for this comment. In general, phylogenetic analyses become more robust if a larger diversity and fuller complement of sequences are included. With very sparse sampling, sequences that are homologous but not orthologous may be misleadingly grouped together, because intermediate sequences have been left out. For tree building, we thus did not want to focus only on experimentally validated receptors but also on all receptors that are phylogenetically related to the validated receptors. Only this approach can ensure a comprehensive exploration of the relationship of peptidergic receptors. The broader phylogenetic approach was also essential to identify orthologs to the experimentally validated Nematostella receptors across other cnidarian species.

(2) Clearly, other neuropeptide signaling systems in cnidarians remain to be discovered but this paper represents a huge step forward.

We appreciate this assessment of the paper. We agree that many systems remain to be discovered. Our paper will also help with the identification of further receptors both in Nematostella as well as other cnidarian species. Please note that we have made specific receptor-ligand predictions for several cnidarian species based on our phylogenetic analysis. Our phylogenies could also help prioritize the study of the remaining orphan Nematostella GPCRs.

(3) There are limitations in what can be interpreted from single cell transcriptomic data but the data nevertheless provide the foundations for future studies involving (i). detailed anatomical analysis of neuropeptide and neuropeptide receptor expression in *N. vectensis* using mRNA in situ hybridization and/or immunohistochemical methods and (ii). functional analysis of the physiological/behavioral roles of neuropeptide signaling systems in *N. vectensis*

We fully agree with this comment. The analysis of the available single-cell sequence resources clearly represents only the first step of anatomical and functional analyses. Our aim was to place the identified peptide-receptor interactions into a whole-organism context with cell type resolution, to highlight the potential complexity of peptidergic signaling in this organism and to facilitate the exploration and conceptualisation of our biochemical screen.

**Comments to authors**
(1) In future, when preparing manuscripts, please use page and line numbers; it makes the task below for reviewers much easier!

We appreciate the suggestion and will do this for future manuscripts.

(2) In the abstract the term "extensively wired" is used. In the context of neuropeptide mediated volume transmission this may not be an appropriate term to use because use of the word "wired" is likely to be associated with point-to-point type classical synaptic transmission; "extensively connected" would be better.

Thank you for this comment. We have changed the text in the abstract to “extensively connected”.

(3) Introduction: Please change "seven-transmembrane proteins and show a slower evolutionary rate than proneuropeptide..." to "seven-transmembrane proteins that show a slower evolutionary rate than proneuropeptide..."

Changed.

(4) Under the section "Creation of a Nematostella neuropeptide library, what is meant by "our regular expressions"? This needs to be rephrased to make it clearer what is meant.

We have now rephrased the relevant sentence to make our approach clearer.

“This predicted secretome was filtered with regular expressions to detect sequences with the repetitive dibasic cleavage sites (K and R in any combination) and amidation sites, using a custom script from a previous publication (Thiel et al., 2021).”

and later:

“Based on the MS data, we included the additional, non-dibasic N-terminal cleavage sites into our script that uses regular expressions to search for repetitive cleavage sites (Thiel et al., 2024) and re-screened the predicted secretome.”

(5) Under the section "Creation of a Nematostella neuropeptide library" the phrase "differ in the length of their N-terminus" needs to be changed to "differ in the length of their N-terminal region". The N-terminus is, as its name implies, one end of the peptide/protein so it can't have a length as such.

Changed.

(6) Under the section "Analysis of metazoan class A GPCRs and selection of *N. vectensis* neuropeptide-receptor candidates",Change:"For a more detailed analysis, we then reduced our sampled species to the cnidarian, the bilaterian with experimentally confirmed GPCRs and Petromyzon marinus, and the two placozoan species (Figure 2B)."To"For a more detailed analysis, we then reduced our sampled species to cnidarians, bilaterians with experimentally confirmed GPCRs and Petromyzon marinus, and two placozoan species (Figure 2B)."

Changed.

(7) Under the section "Analysis of metazoan class A GPCRs and selection of *N. vectensis* neuropeptide-receptor candidates" - change "We re-run" to "We re-ran"

Changed.

(8) Throughout the paper reference is made to a variety of neuropeptides that have or are predicted to have an N-terminal pyroglutmate. However, these are referred to without indicating this post-translational modification e.g. QGRFamide.This should be corrected throughout the paper, in the text, and figures. Two abbreviations for pyroglutamate are used in the literature:pQ, which shows that the encoded amino acid is Q (Glutamine)pE, which shows that the post-translationally modified amino-acid is glutamate (E)In the neuropeptide field, pQ seems to be more widely used than pE, so our recommendation would be to use pQ.

In the revised version we now write pyroQ whenever we refer to the actual peptide. We now only use the peptide name without indicating this modification when we refer to the precursor of these peptides.

(9) The title for Figure 5 is rather short and vague. A title like "Tissue-specific expression of neuropeptide precursors and receptors in Nematostella" seems more appropriate

We appreciate the reviewer's input, and we have made the change accordingly. The revised figure legend now reads: “Tissue-specific expression of neuropeptide precursors and receptors (GPCRs) in *N. vectensis*.”

(10) All of the figures in the paper have been saved in bitmap format (e.g. tiff), which means that the resolution of the figures may end up being poor in the published article. All of the figures in this paper should be saved in vector format (e.g. eps) so that there is no loss of resolution when the size of the file/figure is reduced.

We have now uploaded all figures in vector format (.eps or .pdf) to prevent any loss of resolution.

(11) In Figure 3 - supplement 2 - the neuropeptides are referred to here as PRGamides and GPRGamides. Some consistency is needed here. And in Figure B, the G of one of the GPRGamides is not shown in black.

Thank you for spotting this mistake. We now give the correct peptide sequence in parenthesis as "GPRGamide". We also highlighted the missing GPRGamide in the figure.